# A Novel Global Gridded Ocean Oxygen Product Derived from a Neural Network Emulator and In-Situ Observations

Said Ouala[1], Oussama Hidaoui[2], and Zouhair Lachkar[3]

[1]IMT Atlantique, CNRS UMR Lab-STICC, INRIA Team Odyssey, Brest, France
[2]African Institute for Mathematical Sciences, South Africa
[3]Mubadala Arabian Center for Climate and Environmental Sciences, New York University Abu Dhabi, Abu Dhabi, United Arab Emirates

**Correspondence:** Said Ouala (said.ouala@imt-atlantique.fr)

**Abstract.**

Ocean deoxygenation, driven by climate change, poses significant challenges to marine ecosystems and can profoundly alter nutrient and carbon cycling. Quantifying the rate and regional patterns of deoxygenation relies on spatio-temporal interpolation tools to fill gaps in observational coverage of dissolved oxygen. However, this task is challenging due to the sparsity of observations, and classical interpolation methods often lead to high uncertainty and biases, typically underestimating long-term deoxygenation trends. In this work, we develop a novel gridded dissolved oxygen product by integrating direct oxygen observations with machine-learning-based emulated oxygen estimates derived from temperature and salinity profiles. The gridded product is then generated through optimal interpolation of both the observed and emulated data. The resulting product shows strong agreement with baseline climatology and captures well-known patterns of seasonal variability and long-term deoxygenation trends. It also outperforms current state-of-the-art products by more accurately capturing dissolved oxygen variability at synoptic and decadal scales, and by reducing uncertainty around long-term changes. This study highlights the potential of combining machine learning with classical interpolation methods to generate improved gridded biogeochemical products, enhancing our ability to study and understand ocean biogeochemical processes and their variability under a changing climate.

## 1 Introduction

The global oxygen content of the ocean has been declining over recent decades (Ito et al., 2017; Schmidtko et al., 2017b) and is projected to continue decreasing throughout the current century (Bopp et al., 2013; Kwiatkowski et al., 2020), leading to detrimental consequences for marine organisms (Rabalais et al., 2002; Vaquer-Sunyer and Duarte, 2008; Laffoley and Baxter, 2019) and profound changes in biogeochemical cycles (Codispoti et al., 2001). These changes can affect the ocean's emissions and uptake of greenhouse gases, thereby influencing Earth's climate (Gruber, 2008; Keeling et al., 2010; Lachkar et al., 2024). Recent estimates of the global decline in oceanic oxygen range from 0.5% to 3.3 % relative to the climatology for the period 1970–2010 (Schmidtko et al., 2017b). However, these estimates are subject to significant uncertainty, particularly in data-sparse

regions. A major source of uncertainty in assessing and understanding ocean deoxygenation is the limited spatial and temporal coverage of dissolved oxygen observations.

Despite advances in autonomous profiling floats, underwater vehicles, and large-scale ocean sensing programs such as ARGO, dissolved oxygen observations remain insufficient to accurately estimate deoxygenation rates at both global and regional scales (Gruber et al., 2010; Claustre et al., 2020). Large regions, particularly during the pre-ARGO era, in the South Pacific, the Indian Ocean, and the polar regions remain undersampled (Hermes et al., 2019; Grégoire et al., 2021) and the presence of seasonal biases and irregular sampling, especially before the ARGO era, significantly limits the ability to directly

analyze fine-scale spatio-temporal variability from observations. In this context, assessing global and regional trends in ocean oxygen content requires developing interpolation methods that map available data onto a regular space-time grid. Gridded oxygen products also play a crucial role in validating ocean models, including both global models used in Earth System Models (ESMs) and regional models necessary for projecting the impact of climate change on oxygen at regional scales. However, as highlighted in Ito et al. (2024b), standard interpolation techniques commonly used to generate these products tend to underes-

timate oxygen trends in data-sparse regions, leading to a potential underestimation of global ocean deoxygenation.

Recently, various studies have demonstrated that machine learning (ML) techniques can outperform classical state-of-the-art methods in geosciences for applications such as weather forecasting (Lam et al., 2023; Bi et al., 2023), simulation (Nguyen et al., 2023; Dheeshjith et al., 2024; Ouala et al., 2020, 2023), and data assimilation (Ouala et al., 2018; Cheng et al., 2023). From a modeling perspective, ML techniques have been successfully used to emulate ocean models at both short (Wang et al.,

2024; Aouni et al., 2024) and long timescales (Dheeshjith et al., 2024). Additionally, ML models show promise in developing data-driven, automated tuning methods for ESMs (Ouala et al., 2024; Kochkov et al., 2024), where gridded oxygen products serve as valuable references for model calibration (Sharp et al., 2022; Ito et al., 2024a). ML has also been applied to produce gridded products of ocean variables from partial and noisy ocean observations (Martin et al., 2023, 2024). In this context, recent studies have begun exploring ML-based approaches for generating gridded ocean oxygen products (Sharp et al., 2022;

Ito et al., 2024a). While these studies highlight the potential of ML-based emulators in generating gridded oxygen products, they typically rely on existing gridded temperature and salinity datasets. For instance, Sharp et al. (2022) derived oxygen fields from the (RG09) Argo Climatology (Roemmich and Gilson, 2009), which inherently restricts its applicability to the ARGO era, limiting its use for long-term climate trend analysis. Similarly, Ito et al. (2024a) emulated oxygen using monthly gridded temperature and salinity datasets, obtained from the Hadley Center EN version 4 (Good et al., 2013), extending the temporal

coverage at the cost of excluding marginal seas. Moreover, interpolation errors inherent to these products can bias ML-based emulators and degrade the quality of the data they are trained on, particularly in data-sparse regions.

Here, we propose a novel method to generate a gridded oxygen concentration product covering the period 1965-2022. This time span is comparable to that used in other classical (e.g., Ito et al., 2017; Schmidtko et al., 2017a; Ito, 2022) and ML-based (e.g., Ito et al., 2024a) $O_2$ reconstructions. The product is derived from observed ocean oxygen data combined with

emulated profiles based on temperature and salinity measurements. Specifically, we expand the available oxygen observations by training a neural network emulator to predict oxygen concentrations from temperature and salinity profiles. After a quality control process for the emulated profiles, we apply optimal interpolation (OI) to combine the real and emulated data into a

unified gridded product. This approach offers several advantages over traditional interpolation methods and recent ML-based techniques, including: (i) an increased density of observations, (ii) a stand-alone data product that is independent of existing interpolated products, and (iii) a flexible temporal and spatial resolution that can be adjusted based on the density of available observations. The resulting product (Ouala et al., 2025) demonstrates good agreement with expected spatial and temporal variability of dissolved oxygen, particulalry regarding global deoxygenation rates. It also exhibits improved performance over current state-of-the-art products in better resolving dissolved oxygen variability at synoptic scales of the order of $O(10^3)$km and in reproducing the climatological seasonal cycle near the ocean surface. Interestingly, when compared to classical optimal interpolation of direct dissolved oxygen measurements, our product better resolves decadal variability and significantly reduces the uncertainty of the reconstructed field, particularly in data-sparse regions, where direct measurements of dissolved oxygen are lacking.

## 2    Materials and Methods

We construct a gridded dissolved oxygen concentration product by combining dissolved oxygen observations with emulated estimates derived from temperature and salinity profiles. The process involves:

- **Building quality-controlled datasets:** The quality control (QC) is based on both World Ocean Database (WOD) flags and additional relevant QC criteria inspired by the work of (Schmidtko et al., 2017b).

- **Neural Network Emulation**: A neural network is trained to emulate oxygen profiles from temperature and salinity measurements.

- **Optimal Interpolation:** Observed and emulated data are combined using OI to produce the final gridded product.

Each of these steps is detailed in the following sections.

### 2.1    Data

The data used in this study was obtained from WOD, the largest publicly available collection of uniformly formatted, quality-controlled ocean profile data. We use dissolved oxygen (DO), temperature ($T$), and salinity ($S$) data sourced from various platforms and institutions (refer to Tables C1 and C2 in the SI Appendix C for a detailed description of data sources).

Standard QC from WOD is applied to the $T$, $S$, and DO profiles, retaining only observations flagged as accepted values with flag 0. Furthermore, additional QC is implemented for the oxygen data following the methodology outlined in Schmidtko et al. (2017b). Specifically, oxygen profiles with a maximum-minimum difference of less than 5 $\mu mol/kg$ and those with differences of less than 0.5 $\mu mol/kg$ across 18 depth levels were excluded. Furthermore, profiles with surface oxygen concentrations below 100 $\mu mol/kg$ were removed. Additional quality control steps targeted supersaturation anomalies. Specifically, profiles exhibiting supersaturation at depths exceeding 200 $m$ with supersaturation levels above 115% were discarded. Additionally, profiles with surface oxygen concentrations below 90% of the expected saturation level were excluded.

We constructed two distinct datasets: one for training the ML model and another for generating the gridded DO product.

The dataset used to train the ML model consists of collocated pairs of $T$, $S$ and (DO) data from 1965 to 2022. The dataset was divided into training, validation, and test subsets. The test set comprises 23 independent $1° \times 1°$ regions, distributed across all major ocean basins, including the North Equatorial and South Pacific, North and South Atlantic, and the Indian Ocean. Within each basin, we selected test locations that capture a diversity of oxygen dynamics, including both oxygen minimum zones (e.g., Regions F, G, and H in the Indian Ocean, Region D in the Atlantic, and Regions A, B, and C in the Pacific) and highly oxygenated regions (e.g., Regions P, Q, and R). The locations of these test regions, along with the performance of the machine learning-based emulator, are shown in Figure 3. The remaining data were allocated to training and validation, with 80% used for training and the remaining 20% for validation.

The dataset used to generate the gridded DO product consists of DO data from the preprocessed WOD profiles and emulated DO samples generated by the neural network. To ensure that the emulated DO samples used in the interpolation of the final gridded product fall within the observed range of ocean oxygen variability, the emulated DO data undergo a final QC check, as described in Section 2.3. We highlight in Figure 1 the number of the emulated and real data retained after the QC, with percentage of each data type given in appendix D Figure D4. Overall, adding the emulated profiles significantly improves the sampling of DO in all major ocean basins and reduces the seasonal bias, particularly in the ARGO-era after year 2000. This is further highlighted when comparing the spatial coverage of the observed dissolved oxygen data (Figure D5 in appendix D) and that of the emulated oxygen data (Figure D6 in appendix D).

## 2.2 Machine learning algorithm training and validation

### 2.2.1 Model architecture

The machine learning model used to emulate DO data is a Multilayer Perception (MLP) model (also referred to as a feedforward or fully connected neural network). This model was used in various studies to predict ocean biogeochemical variables, including nutrient concentrations and carbonate system parameters both globally (Sauzède et al., 2017) and in regional configurations (Fourrier et al., 2020). It was also applied to construct gridded oxygen datasets based on interpolated temperature and salinity products (Sharp et al., 2022; Ito et al., 2024a).

Following the approach of Sauzède et al. (2017), the input of the neural network consists of water temperature (T), salinity (S), depth (d), latitude (LAT), longitude (LON) and month of the year (MOY). The periodicity of LON and MOY were accounted using a sinusoidal encoding of these variables. Further details on the model hyperparameters, training algorithm, evaluation metrics, and feature importance analysis are provided in the Supplementary Information (Appendix A).

### 2.2.2 Performance of the ML model

We use independent test regions to evaluate the ability of the trained neural network model to emulate dissolved oxygen concentrations.

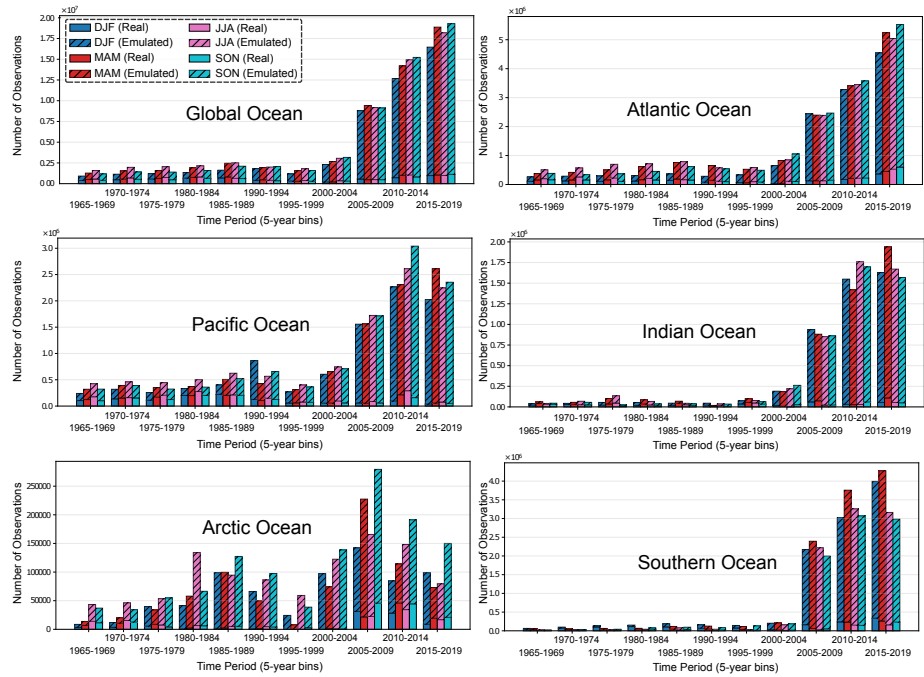

**Figure 1.** *Data sampling by ocean region and season.* Each panel corresponds to a different oceanic region. Solid bars represent real observations, while hatched bars indicate the contribution of *emulated* observations.

To evaluate the epistemic uncertainty of the machine learning model (Valdenegro-Toro and Mori, 2022), we start by training a $k$-fold ensemble of MLP models (with $k = 5$), each one based on different training and validation datasets. The trained models are then used to predict oxygen data in the test set. The results of this experiment are depicted in appendix D, Figure D1, where we visualize the distribution of the standard deviation of the $k$-fold ML ensemble predictions across water layers from the surface down to $2000 \ m$. Overall, the ensemble standard deviations are generally low, with median uncertainties within each depth layer around $5 \ \mu\mathrm{mol\,kg}^{-1}$. This emphasizes that the training of the ML model is stable and that the $k$-fold training methodology is able to recover some of the epistemic uncertainty (Valdenegro-Toro and Mori, 2022) related to limitations in the data coverage and/or model parameterization. However, using this uncertainty as the error estimate for the emulated profiles in the optimal interpolation would have made the interpolation overconfident in the emulated profiles relative to the real observations. Therefore, we use only a single MLP model (the best-performing one) for emulating the profiles used in the interpolation. This model is further evaluated in the subsequent analyses. The associated error estimate of this model, described in Section 2.4.3, is set higher than that of the real observations.

The scatter plot of the predicted vs. true oxygen values highlighted in Fig. 2 shows that the trained model is able to accurately predict dissolved oxygen concentration with a root mean squared error $RMSE = 12.13 \mu mol/kg$ and a correlation coefficient $R^2 = 0.98$.

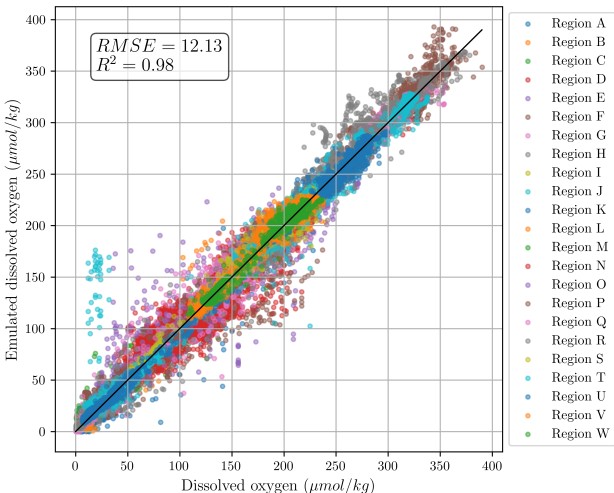

**Figure 2.** *Scatter plot of the emulated oxygen data with respect to the ground truth measurements.* The plot includes emulated oxygen values across all depths and within all independent test regions. The locations of the independent test regions is given in Figure 3.

An analysis of the average vertical profiles based on observations and as emulated by the ML model indicates that the model aligns well with the observed profile distribution and successfully captures dissolved oxygen variability in all independent test regions (Figure 3). In particular, the model accurately predicts low oxygen concentrations associated with Oxygen Minimum Zones (OMZs), as well as the vertical and spatial variability of oxygen in test regions located far from OMZs.

To identify potential trends and biases in the neural network model, we further evaluate the distribution of the difference between the emulated oxygen values and the observations in the independent test regions, as shown in Figure 4. Overall, the boxplots indicate that the model accurately captures the variability of ocean oxygen across both space and time. Notably, larger errors are observed in the upper ocean layers, which correspond to higher oxygen concentrations. Additionally, higher errors are seen in the earlier years, likely due to the limited number of observations during this period.

## 2.3 Validation of oxygen data emulated by machine learning

The trained ML model can generate outliers, particularly when emulating oxygen data based on temperature and salinity measurements at locations that were not represented in the training data. This phenomenon, known as out-of-distribution sampling in the ML community, can result in biases in the final interpolated product, especially in regions with a high density of emulated data but few or no actual ocean oxygen observations.

To address this issue, we design a QC framework to validate the emulated oxygen data before incorporation into the global interpolation. The primary goal of this framework is to exclude anomalous emulated oxygen data while minimizing the rejection of valid data. The QC framework begins by computing validity thresholds, derived from historical dissolved oxygen measurements. These thresholds represent the minimum (min) and maximum (max) observed values, calculated for each loca-

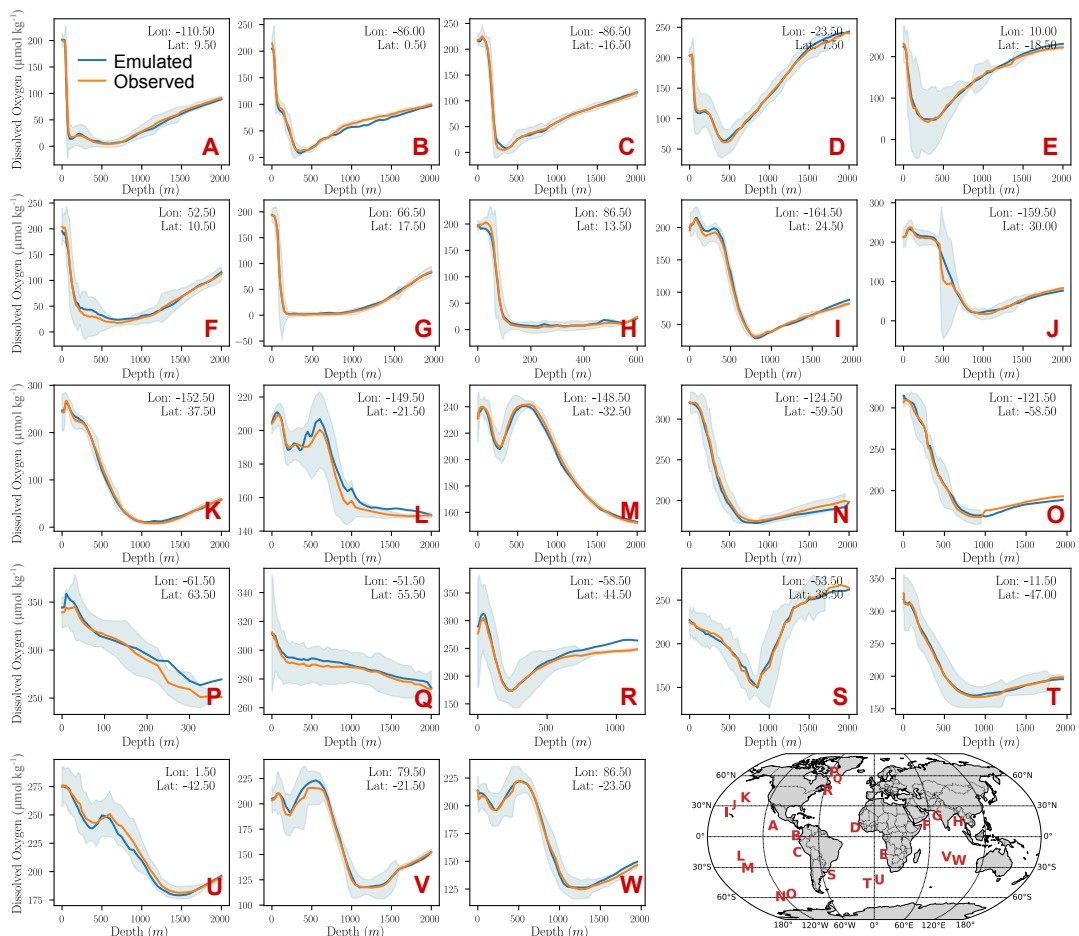

**Figure 3.** *Mean emulated oxygen profiles compared to the empirical distribution of observed oxygen profiles in the independent test regions.* The mean emulated profile (orange) against the empirical distribution of observed dissolved oxygen profiles (blue line and shading) in the independent $1° \times 1°$ test regions. The longitude and latitude of each of the 23 test regions are indicated in the corresponding panels. The locations of the test regions are also shown in the bottom-right corner of the figure.

tion and depth of the interpolation grid. They reflect the natural variability of oxygen in the ocean and serve as a baseline for identifying potential outliers (please refer to Figure D2 in the SI Appendix D for an illustration of these min/max filters.)

The computation of these min-max fields is based on a dynamic binning of the observations in the interpolation grid, where the measurements used to compute the min-max statistics of each grid cell can include neighboring grid cells until a sufficient number of 15 observations is collected. The observations are also aggregated within a $\pm 5$-year window around the target year. This temporal aggregation helps increasing the number of observation points in data-poor regions and reduces the impact of short-term anomalies while capturing broader trends in oxygen variability.

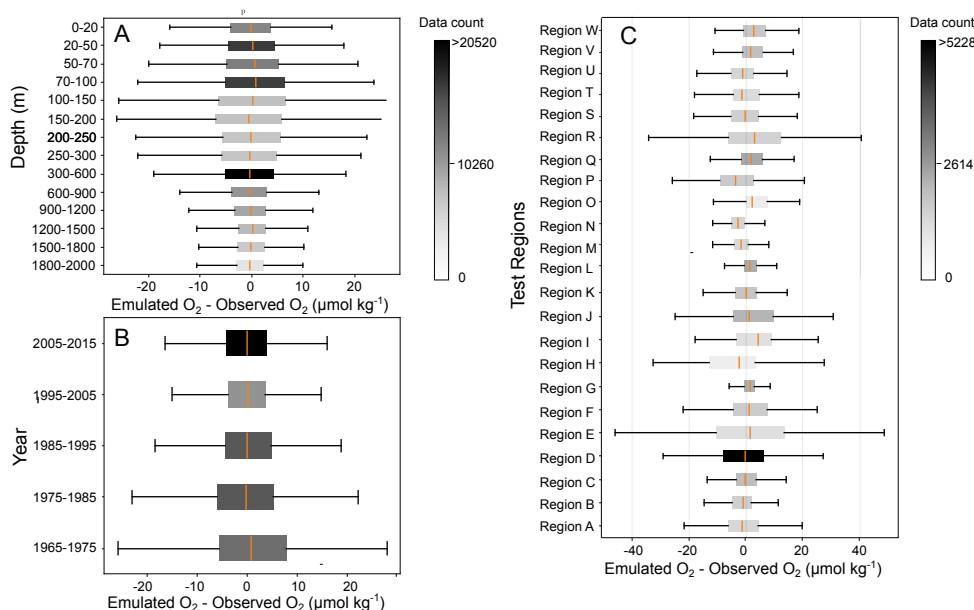

**Figure 4.** *Box plots of the differences between emulated and observed oxygen concentrations in the independent test regions.* The box plots are binned based on depth, year and location of the test region. The intensity of the shading in each box refers to the number of data points. For each box, the negative and positive whiskers represent the Q1–1.5*IQR and Q3 + 1.5*IQR, respectively, where Q1 is the 0.25 quantile, Q3 the 0.75 quantile, and IQR the inter-quantile range. The width of each box represents the IQR and the middle line the median of the values.

Emulated oxygen values are validated by comparing them with the established thresholds for their specific location, depth, and time. Values falling outside the range defined by the min and max thresholds are flagged as outliers and excluded from the interpolation.

## 2.4 Optimal Interpolation

We perform an interpolation of both observed and emulated dissolved oxygen data using a standard Optimal Interpolation (OI) method. Following the substantial increase in temperature and salinity data coverage in the ARGO era (after 2002), which also corresponds to an increase in emulated oxygen profiles, we construct two gridded products with different temporal resolutions. The first product, with yearly resolution from 1965 to 2022, is designed to study decadal and climate-change-related variability. The second product, with a monthly resolution, focuses on the ARGO era (2005-2022) and aims to capture seasonal and interannual variability. Both products use a $1°$ horizontal resolution, which is standard among most objectively analyzed products based on in-situ data (Cheng et al., 2017; Ito, 2022), including ML-based reconstructions (Sharp et al., 2022; Ito et al., 2024a). The vertical grid comprises 65 standard WOD depth levels ranging from 0 m to 2000 m, matching the typical vertical extent of most ARGO flaots (Roemmich et al., 2009).

### 2.4.1 General description

Quality-controlled dissolved oxygen data and emulated data are binned into a global $1° \times 1°$ horizontal grid with 65 standard WOD depth levels ranging from 0 m to 2000 m. These binned data are interpolated using standard Optimal Interpolation (OI) to produce an analysis field of dissolved ocean oxygen and its corresponding uncertainty.

The OI method combines a background information with observations to derive the analysis state as follows:

$$\mathbf{x}_a = \mathbf{x}_b + \mathbf{K}(\mathbf{y} - \mathbf{H}\mathbf{x}_b) \tag{1a}$$

$$\boldsymbol{\Sigma}_a = \boldsymbol{\Sigma}_b - \mathbf{K}\mathbf{H}\boldsymbol{\Sigma}_b \tag{1b}$$

where $\mathbf{x}_a$ is the analyzed field, $\boldsymbol{\Sigma}_a$ is the covariance of the analysis, $\mathbf{x}_b$ is the background field, $\boldsymbol{\Sigma}_b$ is the covariance of the background field, $\mathbf{y}$ are the binned observations, and $\mathbf{H}$ is the observation operator. The Kalman gain $\mathbf{K}$ is computed as:

$$\mathbf{K} = \boldsymbol{\Sigma}_b \mathbf{H}^\top \left(\mathbf{H}\boldsymbol{\Sigma}_b\mathbf{H}^\top + \boldsymbol{\Sigma}_o\right)^{-1}, \tag{2}$$

where $\boldsymbol{\Sigma_o}$ are the background and observation error covariance matrices, respectively.

### 2.4.2 Background and observation error covariances

For convenience, we drop the dependence of the Kalman Gain matrix on the observation model by introducing the data-grid and the data-data covariance matrices $\boldsymbol{\Sigma_{xy}}$ and $\boldsymbol{\Sigma_{yy}}$ respectively which are defined as follows:

$$\boldsymbol{\Sigma_{xy}} = \boldsymbol{\Sigma}_b \mathbf{H}^\top \tag{3a}$$

$$\boldsymbol{\Sigma_{yy}} = \mathbf{H}\boldsymbol{\Sigma}_b\mathbf{H}^\top \tag{3b}$$

In this work, the covariance matrices $\boldsymbol{\Sigma_{xy}}$ and $\boldsymbol{\Sigma_{yy}}$ are built using the same isotropic Gaussian prior as follows:

$$\boldsymbol{\Sigma_{x\cdot}} = \sigma_b^2 exp(-\frac{L_{m,n}}{2L_h}) \tag{4}$$

where $\sigma_b^2$ is the total variance of the background field, and $L_{m,n}$ is the distance between two grid points $m$ and $n$. $L_h$ is the e-folding horizontal length scale. In this work, we follow the approach of Ito (2022) and set $L_h$ to 1000 km for the interpolation of the yearly product between 1965 and 2022. For the monthly product, a larger number of observations are available, allowing us to reduce $L_h$ to 300 km, which is consistent with ARGO-based products of temperature and salinity fields (Gaillard et al., 2016).

The background covariance matrix is assumed to be diagonal and is computed from the variance of the binned observations.

### 2.4.3 Observation Error Covariance Matrix

The observation error covariance matrix $\boldsymbol{\Sigma}_o$ is assumed to be diagonal. Each diagonal element $(\boldsymbol{\Sigma}_o)_{n,n}$ represents the variance of the binned observations at grid point $n$. This variance is derived from the following sources of uncertainty:

**Gridding Uncertainty** ($\sigma_g^2$): This component arises from approximating the oxygen distribution at each grid cell using the empirical distribution based on the observed and emulated samples.

**Measurement Uncertainty** ($\sigma_m^2$): This source of uncertainty is attributed to each dissolved oxygen measurement. It is estimated, following the methodology described in Sharp et al. (2022), as 3% of the gridded oxygen measurements.

**Emulation Uncertainty** ($\sigma_e^2$): This uncertainty arises from the emulated oxygen data. As discussed in section 2.2.2, the gridded emulated oxygen measurements align with gridded oxygen data. In this context, the uncertainty estimate of the emulated oxygen data is computed similarly to the dissolved oxygen measurements, with a slight increase to 4% of the gridded emulated oxygen measurements.

The total variance at grid point $n$ is computed, assuming independence, as:

$$(\mathbf{\Sigma}_o)_{n,n} = \sigma_{g,n}^2 + \sigma_{m,n}^2 + \sigma_{e,n}^2, \tag{5}$$

### 2.4.4 Data aggregation

The observed and emulated oxygen data are aggregated over a time window based on the product's resolution. For the yearly (respectively, monthly) product, each grid cell at year (respectively, month) $t$ includes data from $t \pm 2$ years (respectively, $t \pm 2$ months).

## 3 Results

We analyze the variability of dissolved oxygen in our gridded oxygen product and compare it with that based on the World Ocean Atlas 23 (WOA 23) and other existing ML-based products (Sharp et al., 2022; Ito et al., 2024a). Specifically, we examine the annual-mean climatology, as well as the seasonal and long-term variability, and contrast our findings with those from these reference datasets.

### 3.1 Annual mean climatology

We first compare the climatological spatial distribution of dissolved oxygen in our product to the WOA23 baseline (Figure 5). The comparison is based on the yearly dataset because of its longer temporal coverage. The horizontal variability of oxygen at a depth of 200 m closely matches the spatial distribution of oxygen concentrations in the WOA23 product, showing strong agreement in the location and intensity of oxygen minimum zones, as well as the high oxygen concentrations at higher latitudes. Similarly, the vertical distribution in the Atlantic, Indian, and Pacific basins aligns well with WOA23, accurately capturing vertical oxygen gradients and the depth and intensity of oxygen minimum zones.

Beyond the climatological spatial distribution, we assess the spatial resolution of the proposed monthly DO product relative to state-of-the-art machine learning-based gridded datasets. Figure 6 presents an example of the DO anomaly field in the equatorial Pacific (-179°E to -100°E, 30°S to 30°N) alongside the vertically and monthly averaged Radially Averaged Power Spectral Density (RAPSD) of our product, compared with GOBAI-O2 and Ito et al. (2024a). This region is characterized by energetic synoptic-scale variability (Chelton et al., 2007), and we evaluate whether our product better captures these processes.

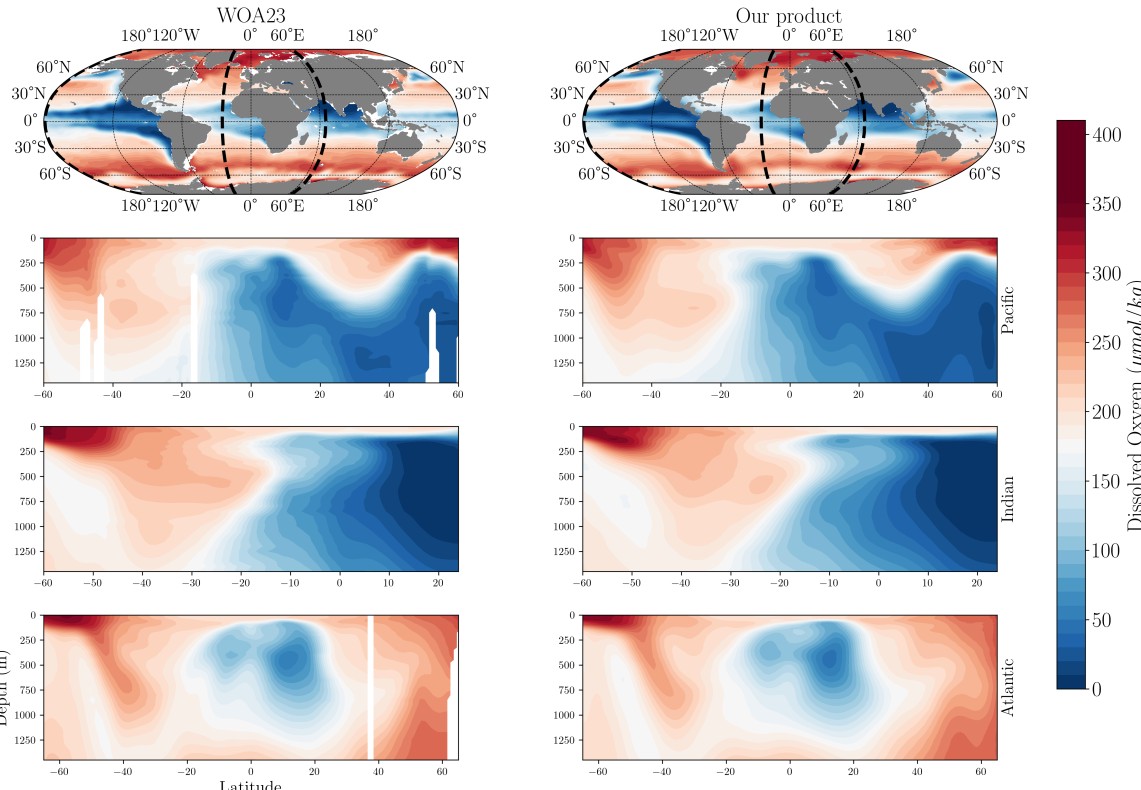

**Figure 5.** *Spatial distribution of dissolved oxygen concentration in our product compared to WOA23.* The first row shows the climatological annual-mean dissolved oxygen concentration at 200 m depth. Dashed black lines represent the locations of the meridional cross sections from the surface to 1500 m. The remaining three rows display oxygen along meridional cross-sections in the Pacific (-179 °E), Indian (65 °E), and Atlantic (-25 °E) basins in WOA23 (left) and our product (right).

The RAPSD indicates 100-200% higher energy levels at wavelengths around $O(10^3)$ km, compared to the ML baselines, suggesting an improved representation of small-scale variability. This is further illustrated through the visual analysis of the anomaly field in Figure 6, where our product better represents finer synoptic-scale structures on the order of $O(10^3)$ km, revealing more energetic small-scale eddies than the other ML-based products. These variations are unlikely to result from interpolation biases associated with data scarcity, which would instead tend to produce near-zero anomalies. Likewise, the possibility that they arise from spurious or low-quality profiles is also unlikely, as similar patterns are consistently observed across the other ML-based products.

### 3.2 Seasonal variability

We also compare the climatological seasonal cycle of oxygen in our product across both hemispheres with that from WOA23, as well as with the GOBAI-O2 product (Sharp et al., 2022) and the Ito et al. (2024a) product (Figure 7). Overall, the seasonal

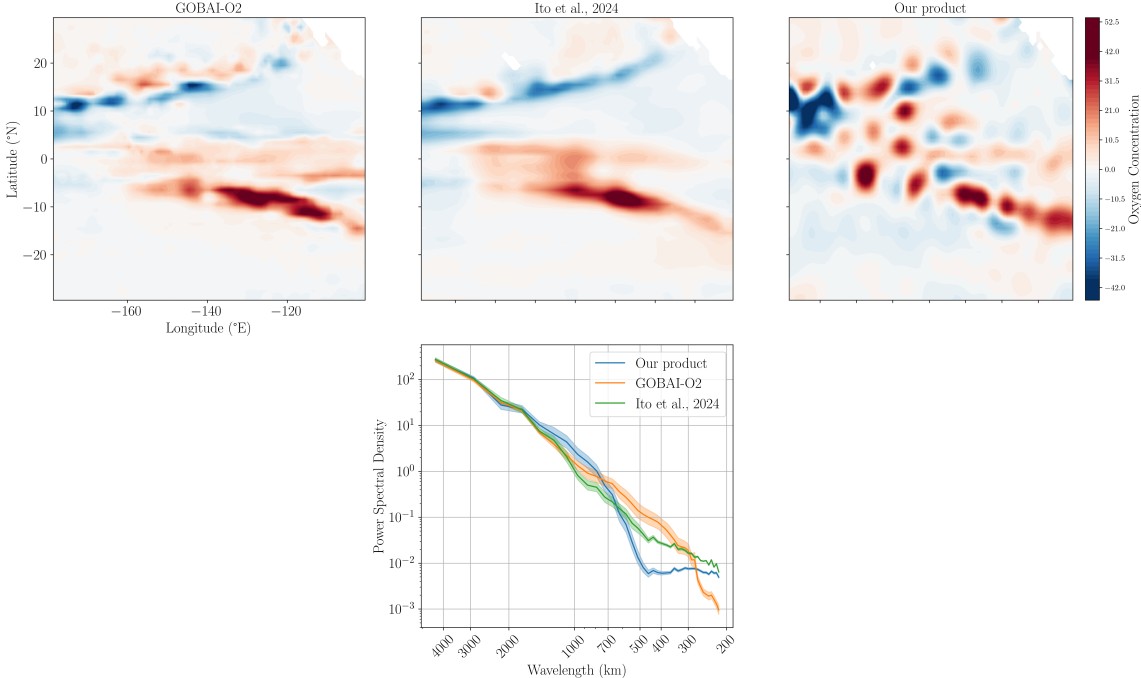

**Figure 6.** *Snapshot of oxygen monthly anomalies in the equatorial Pacific.* Snapshot of the anomaly of the gridded product in the equatorial Pacific region (-179°E to -100°E, 30°S to 30°N) in July 2015 compared to the GOBAI-O2 and Ito et al. (2024a)'s ML baselines. The upper row displays a horizontal section at 200 m depth. The bottom panel shows the distribution of the vertically and monthly averaged RAPSD computed in the equatorial Pacific region.

oxygen cycle in our product closely aligns with WOA23 and previous ML-based estimates. As expected, Figure 7 shows that oxygen seasonality is more pronounced in the upper ocean, reflecting the strong seasonal variability in oxygen saturation, which is primarily driven by temperature variations. This relationship is further quantified by the Pearson correlation coefficients between dissolved oxygen and temperature anomalies reported in Table 1.

Indeed, when compared to GOBAI-O2 and Ito et al. (2024a), our product shows a stronger correlation between upper-ocean
oxygen seasonality and temperature variability, particularly in the Southern Hemisphere. This relationship is more pronounced relative to previous ML-based products.

### 3.3 Long-term oxygen changes and comparison with previous estimates

We analyze long-term changes in ocean oxygen levels using our yearly product and compare them to estimates from ML-based reconstructions by GOBAI-O2 and Ito et al. (2024a) (Figure 8). To ensure a consistent comparison, we use a common ocean
volume across all three products, excluding regions absent in previously published ML datasets. These include marginal seas (not included in Ito et al. (2024a) and the Arctic (>80°N) and Antarctic (<80°S) oceans (not included in GOBAI-O2). Oxygen

| Region | Dataset | Correlation Coefficient |
|---|---|---|
| Northern hemisphere | Our Product | -0.842 |
| | WOA23 | -0.778 |
| | Ito et al., 2024 | -0.476 |
| | GOBAI-O2 | **-0.852** |
| Southern hemisphere | Our Product | **-0.935** |
| | WOA23 | -0.756 |
| | Ito et al., 2024 | -0.835 |
| | GOBAI-O2 | -0.890 |

**Table 1.** *Correlation coefficients between the seasonal anomalies of oxygen and temperature in the upper ocean.* Correlation coefficients are computed for the upper $100m$ with respect to seasonal anomalies in temperature and oxygen in both the Northern and Southern hemispheres.

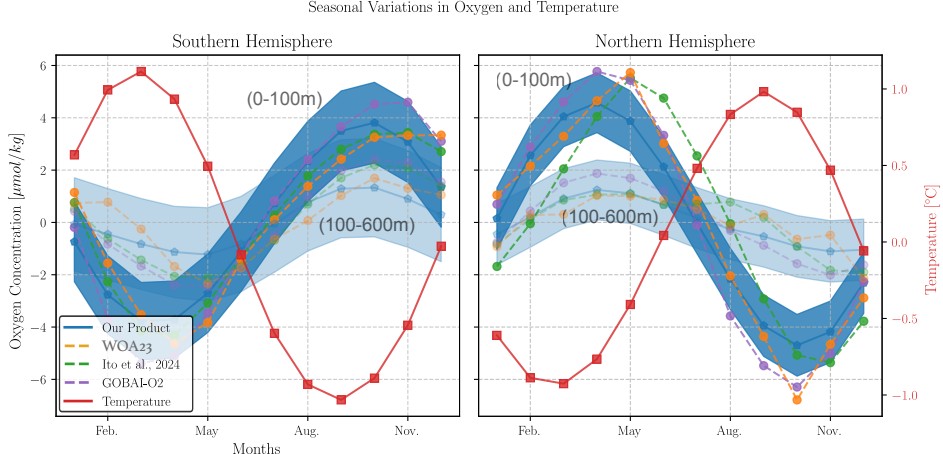

**Figure 7.** *Climatological seasonal cycle in dissolved oxygen in our product compared to WOA23 and previous ML-based models.* The seasonal cycle of dissolved oxygen in our product is shown for the upper 100 m (dark blue) and the 100–600 m subsurface layer (light blue) in the Southern (left) and Northern (right) Hemispheres. Shaded envelopes represent the uncertainty range around the seasonal means. For comparison, the seasonal oxygen cycles from WOA23 (orange) and previously published ML-based reconstructions by Ito et al. (2024a) (green) and GOBAI-O2 (purple) are also displayed. Additionally, the seasonal cycle of temperature in the upper 100 m from WOA23 is shown in red.

content computed based on the entire ocean volume of the proposed product (including marginal seas and polar regions) are depicted in Figure D7 in SI Appendix D.

Our analysis reveals a deoxygenation rate of approximately $-438.37 \pm 52.19$ Tmol/decade in the upper 1000 m over the period 1965–2022, which closely aligns with the $-452.79$ Tmol/decade reported by Ito et al. (2024a) for the same layer and

a similar time span. When extending the analysis to the upper 2000 m, the deoxygenation rate increases to $-685.53 \pm 113.25$ Tmol/decade.

Over the past two decades (2004–2022), our product suggests a slower deoxygenation rate, estimated at $-296.54 \pm 111.37$ Tmol/decade in the upper 1000 m and $-403.73 \pm 193.85$ Tmol/decade in the upper 2000 m. These values are more conservative than those from GOBAI-O2, which reports $-384.48$ Tmol/decade and $-720.99$ Tmol/decade for the same depth ranges and time period. When including marginal seas and polar regions, our estimated deoxygenation rates for the full 1965–2022 period increase to $-582.40 \pm 80.63$ Tmol/decade in the upper 1200 m and $-803.36 \pm 147.20$ Tmol/decade in the upper 2000 m (Figure D7 in SI Appendix D).

We also compare our deoxygenation estimates with previous results based on traditional mapping techniques. In the upper 2000 m, our estimated deoxygenation rate is comparable to the $-960.4 \pm 409.1$ Tmol/decade reported by Schmidtko et al. (2017b) for the period 1965–2010. However, when considering only the upper 1200 m, our estimate suggests a significantly faster deoxygenation rate than Schmidtko et al. (2017a)'s estimate of $-257.5 \pm 185.1$ Tmol/decade over the same period.

Nonetheless, our results exhibit better agreement with Schmidtko et al. (2017a) when analyzed over the same time span (1965–2010), yielding a deoxygenation rate of $-478.06 \pm 125.51$ Tmol/decade in the upper 1200 m and $-644.12 \pm 227.64$ Tmol/decade in the upper 2000 m.

Additionally, we compare our results with estimates from Ito (2022), which were derived using optimal interpolation. Our findings indicate a substantially faster deoxygenation trend. Specifically, we estimate a rate of $-340.84 \pm 151.27$ Tmol/decade in the upper 700 m and $-763.11 \pm 184.74$ Tmol/decade in the upper 2000 m, whereas Ito et al. (2022) reported significantly lower rates of $-100$ Tmol/decade in the upper 700 m and $-327 \pm 45.99$ Tmol/decade in the upper 2000 m. It is known that the estimates from Ito et al. (2022) tend to be lower compared to other state-of-the-art studies (Schmidtko et al., 2017a). Recent research has shown that optimal interpolation methods can introduce biases, particularly underestimating deoxygenation in regions with sparse observational data (Ito et al., 2024b). In this context, our methodology mitigates some of these biases by increasing the density of dissolved oxygen data through the incorporation of ML-emulated oxygen estimates.

Finally, it is worth noting that our product reveals a much stronger decadal and inter-decadal variability in the rate of deoxygenation compared to previous ML-based reconstructions (Figure 8). The influence of decadal climate variability on regional and global deoxygenation is well established (Oschlies et al., 2018). For instance, the rate of deoxygenation in our product from 1980 to the early 2000s was substantially higher than during the 1960s and 1970s, as well as over the past decade. These variations, largely absent in earlier ML-based reconstructions, are consistent with model-based studies suggesting that major climate variability modes, such as the Pacific Decadal Oscillation (PDO), strongly influence ocean oxygen content (Deutsch et al., 2011; Duteil et al., 2018; Ito et al., 2019). For example, Deutsch et al. (2011) showed that PDO explains about 24% of the variability in the volume of suboxic waters in the Pacific based on a model simulation, attributing this relationship to PDO-driven modulation of trade winds, thermocline depth, and respiration rates in the eastern tropical Pacific. Duteil et al. (2018) demonstrated that the sluggish equatorial circulation during positive PDO phases (such as in the 1980s and 1990s) results in a pronounced deoxygenation in the eastern equatorial Pacific and an intensification of its OMZ. Ito et al. (2019) further emphasized the importance of PDO-driven vertical displacements of isopycnals in modulating tropical Pacific ocean

oxygen content. More recently, Poupon et al. (2023) showed that deoxygenation is favored during positive phases of the PDO, whereas negative PDO phases, dominant in the 1960s, 1970s, and over the past two decades, enhance oxygenation in the tropical Pacific, thereby partly offsetting anthropogenic or climate change–driven deoxygenation. Importantly, these decadal oxygen variations are unlikely to arise from interpolation biases related to data scarcity. According to previous work by Ito et al. (2024b), such biases would have tended to produce weaker deoxygenation rates during the pre-ARGO period of limited observations compared to the ARGO period, when observation density increased nearly tenfold. However, in our product, the deoxygenation rate is actually weaker during the ARGO period (2003–2022) than in the pre-ARGO period (1980–2000).

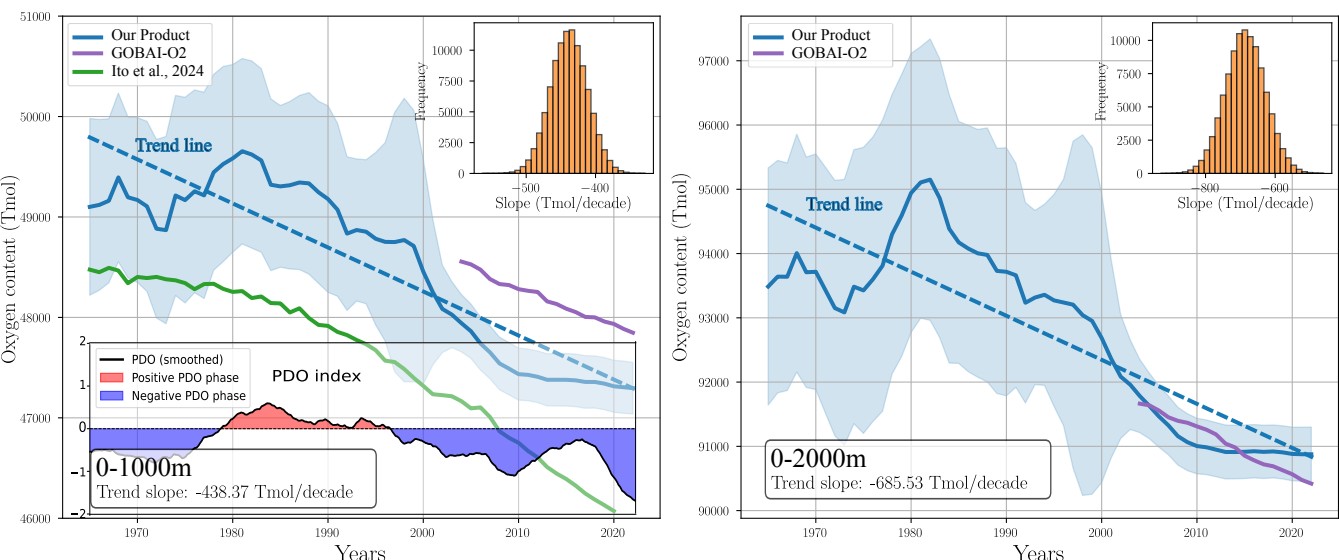

**Figure 8.** *Long-term changes in globally integrated oxygen inventory in our product and other ML-based products.* The left panel shows the inventory for the upper 1000 m, while the right panel displays the inventory for the upper 2000 m. The histograms in each panel represent the empirical distribution of the deoxygenation rate, computed from a Monte Carlo simulation with 10,000 realizations. For comparison, changes in oxygen inventories based on previously published ML-based reconstructions by Ito et al. (2024a)(green) and GOBAI-O2 (purple) are also displayed.

### 3.4 Uncertainty estimates

We analyze the uncertainty fields associated with the proposed gridded product. Uncertainty is quantified using the covariance matrix $\Sigma_a$ of the reconstructed field. As described in Section 2.4, $\Sigma_a$ is diagonal, with each entry representing the variance of the estimated value at a specific grid point after assimilating the data. This variance reflects the remaining uncertainty at that location, accounting for both the background (prior) variance and the information provided by the observations. The variance at a given grid point decreases as the number of nearby observations increases and as the quality of those observations improves. In our framework, the quality of the gridded observations is defined through the observation error covariance matrix, which

assigns lower uncertainty to real dissolved oxygen data than to emulated data. Additionally, the gridding variance accounts for the natural variability of the ocean within a grid cell and for potential disagreement between real and emulated profiles, further reflecting the overall uncertainty at each location.

Our methodology substantially expands the effective observational network by emulating dissolved oxygen (DO) profiles from measured temperature and salinity data (Figure 9a). This enhanced data coverage leads to a significant reduction in the uncertainty of the reconstructed field relative to an optimal interpolation based solely on direct DO observations (Figure 9a). The improvement is particularly evident after 2000, which corresponds to the deployment of the Argo program, which significantly improved the global coverage of T/S measurements used here to emulate oxygen data.

This improvement is further evidenced by comparing the optimal interpolation standard deviation when using only the direct observations versus when including the emulated profiles as well (Figure 9b). The baseline product, based solely on direct observations, exhibits high uncertainty, particularly in data-sparse regions. In contrast, the proposed product—which incorporates both direct and emulated profiles—demonstrates significantly reduced uncertainty due to the enhanced spatial coverage.

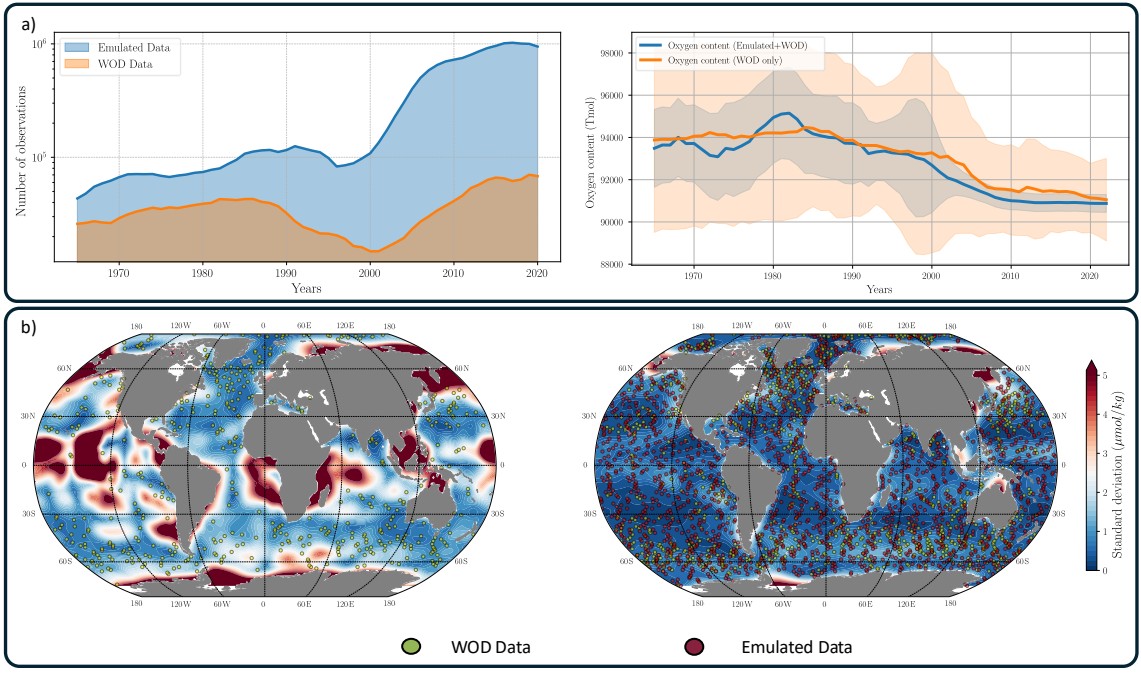

**Figure 9.** *Analysis of uncertainty of the gridded oxygen product.* Panel (a, left) shows the total number of direct DO observations and the number of emulated observations used in the gridding process of the OI at 200 m depth. Panel (a, right) compares the global DO inventory derived from two OI fields: The proposed product (in blue), which takes into account both direct and emulated data, and a baseline field (in orange) that relies exclusively on direct measurements. Panel (b) presents the standard deviation of the OI fields at 200 m depth. The left map corresponds to the OI using only direct observations, while the right includes both direct and emulated profiles. The spatial distribution of direct (green) and emulated (red) observations is displayed, with a subsampling factor of 15 applied to improve visibility.

## 4 Conclusion and outlook

In this study, we developed a novel gridded dissolved oxygen concentration product by integrating observational oxygen data with emulated estimates derived from temperature and salinity profiles. The proposed methodology offers several key advantages, including:

- **Increased Density of Observations:** The methodology significantly increases the density of the observational coverage, which in turn reduces the uncertainty and biases of the gridded product. The improvement is particularly strong in data-poor regions.

- **Independent Interpolation of Oxygen Data:** Unlike recent state-of-the-art ML approaches (Sharp et al., 2022; Ito et al., 2024a) that derive gridded oxygen products from interpolated temperature and salinity fields, our method directly interpolates observed and emulated oxygen data using optimal interpolation (OI). This approach ensures better control over the sources of errors present in data, leading to more interpretable uncertainty estimates of the final gridded product.

- **Flexible Space-Time Resolution:** The approach provides flexibility in the spatial and temporal resolution of the gridded product, which can be adjusted based on the density of available observations. Specifically, we generate two gridded products: one with a yearly resolution covering the entire study period (1965–2022) and another with a monthly resolution from 2005 to 2022, capitalizing on the denser data coverage during the Argo period. Both products are constructed with a 1°resolution. However, the effective resolution of the monthly product is enhanced by using a smaller e-folding length scale in the OI process.

The resulting product generally agrees with the reference climatology and recent ML-based products in terms of reproducing the spatial variability of dissolved oxygen, and it also aligns with previously published estimates of long-term global deoxygenation, albeit with reduced uncertainty around those estimates. Moreover, the product reveals interdecadal variability that is absent from existing ML-based reconstructions but consistent with numerical model simulations, suggesting that it better captures the underlying physical processes at these scales.

The proposed methodology can be extended to other biogeochemical variables, including, for example, nutrients, pH, phytoplankton pigments, and particle backscatter. Previous studies (e.g., Sauzède et al., 2017) have shown that neural network models can predict such variables based on dissolved oxygen measurements. In this context, our study highlights the potential to extend these works by developing biogeochemical emulators based on physical measurements and emulated oxygen data. This would enable the construction of gridded products for these variables with improved spatial and temporal resolutions.

While our results confirm global deoxygenation trends that are consistent with previous studies, further research is needed to investigate regional-scale deoxygenation, particularly in areas where major OMZs are present. In this context, the increased observational density provided by our ML-emulated oxygen data can enhance our ability to monitor and document ongoing changes in major OMZs, while also reducing uncertainties in these estimates.

## 5 Data availability

The gridded products developed in this study are publicly available in (Ouala et al., 2025).

*Author contributions.* SO and ZL designed and structured the study. SO and OH developed the model code and generated the gridded product. SO and ZL analyzed the data and prepared the manuscript.

*Competing interests.* The authors declare that they have no competing interests.

*Acknowledgements.* SO acknowledges the support of the AI4Extremes Chair and the Horizon project AI4PEX (grant agreement 101137682). ZL was supported by funding provided by Tamkeen through grant CG009 to the Mubadala ACCESS Center and funding support from Mubadala Philanthropies under XR016; their support is greatly appreciated. Data processing and machine learning model training was performed at the High Performance Computing (HPC) cluster of NYUAD, Jubail.

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

## Appendix A: Parameterization and training of the ML model

The machine learning model used to emulate dissolved oxygen concentrations is a fully connected neural network with six layers. Its architecture includes an input layer with 10 features (detailed hereafter), four hidden layers with 256, 128, 128, 64, and 32 neurons, respectively, and an output layer with a single neuron.

The input features include water temperature ($T$), salinity ($S$), depth (d) latitude (LAT), longitude (LON), month of the year (MOY), day of the month (DOY), and calendar year (Y). To capture the periodicity of LON and MOY, a sinusoidal encoding is applied:

$$\mathbf{MOY}_s = \begin{bmatrix} \cos\left(\frac{MOY \times \pi}{6}\right) \\ \sin\left(\frac{MOY \times \pi}{6}\right) \end{bmatrix} \quad ; \quad \mathbf{Lon}_s = \begin{bmatrix} \cos\left(\frac{LON \times \pi}{180}\right) \\ \sin\left(\frac{LON \times \pi}{180}\right) \end{bmatrix} \tag{A1}$$

where the subscript $s$ refers to the sinusoidal encoding.

The hidden layers utilize the rectified linear unit activation function (*relu*). The model is trained using the mean squared error (MSE) loss function and is optimized with the Adam optimizer (Kingma and Ba, 2014). The MLP architecture and hyperparameters were selected through incremental testing, starting from a simple configuration and gradually increasing model complexity until no further improvement was observed on the training of the model.

## Appendix B: Feature Importance Analysis

We further investigate the contribution of each input feature to the machine learning model's predictions of dissolved oxygen. To this end, we employ the Integrated Gradients method () and report in Figure B1 the normalized absolute importance of each feature.

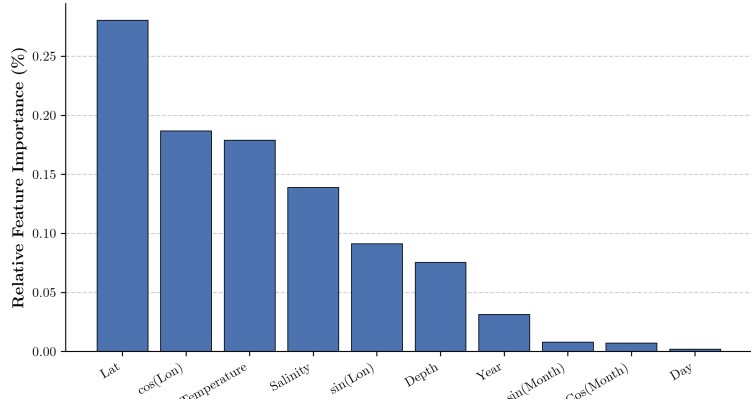

**Figure B1.** *Relative importance of each input feature for predicting dissolved oxygen.*

Overall, the most influential features are the geographical location of the profile (latitude and longitude) and the physical
variables, namely temperature and salinity, while temporal information (year, month, day) has the least influence. Geographical
location serves as a proxy for the various oceanographic regimes of ventilation and biogeochemistry that characterize different
ocean regions (e.g., tropical vs. high-latitude regions, western boundary currents vs. OMZs). Temperature and salinity directly
influence the solubility of oxygen in seawater, as well as stratification, water mass transformations, and mixing. The minimal
contribution of time-related variables (year, month, day) likely reflects the fact that most of their information content—such
as seasonality and the gradual, long-term deoxygenation trend—is already captured by the time-varying physical variables
(temperature and salinity).

## Appendix C: Data sources

This section outlines the data sources used in this study. The data are organized into two distinct databases. The first database,
presented in Table C1, includes data sources and instruments that provide collocated measurements of temperature, salinity,
and dissolved oxygen. This dataset is utilized for training, validation, and testing of the neural network emulator. Oxygen data
from this dataset are also used in the optimal interpolation.

| WOD dataset | Instrument Type |
| --- | --- |
| OSD | Bottle, low-resolution CTD, low-resolution XCTD, plankton data |
| CTD | High-resolution CTD and high-resolution XCTD data |
| PFL | Profiling float data, mainly from the Argo program |
| DRB | Drifting buoy data from surface drifting buoys with thermistor chains and ice-tethered profilers |
| UOR | Undulating Oceanographic Recorder data from a CTD probe on a towed vehicle |
| GLD | Glider data |
| MRB | Moored buoy data, mainly from the Equatorial buoy arrays (TAO) |

**Table C1.** ***Data sources used to construct the training, validation and testing dataset of the ML-based emulator.*** Data access: 6/12/2023.

The second database, presented in Table C2, consists of only temperature and salinity data. This dataset is used to generate
emulated oxygen data using the trained ML emulator.

## Appendix D: Additional figures

| WOD dataset | Instrument Type |
| --- | --- |
| OSD | Bottle, low-resolution CTD, low-resolution XCTD, plankton data |
| CTD | High-resolution CTD and high-resolution XCTD data |
| PFL | Profiling float data, mainly from the Argo program |
| DRB | Drifting buoy data from surface drifting buoys with thermistor chains and ice-tethered profilers |
| UOR | Undulating Oceanographic Recorder data from a CTD probe on a towed vehicle |
| GLD | Glider data |
| MRB | Moored buoy data, mainly from the Equatorial buoy arrays (TAO) |
| SUR | Surface-only data (bucket, thermosalinograph) |
| APB | Autonomous Pinniped Bathythermograph - TT-D recorders and CTDs attached to elephant seals |

**Table C2.** *Data sources used to construct the temperature and salinity dataset used to generate the emulated oxygen data.* Data access: 15/5/2024.

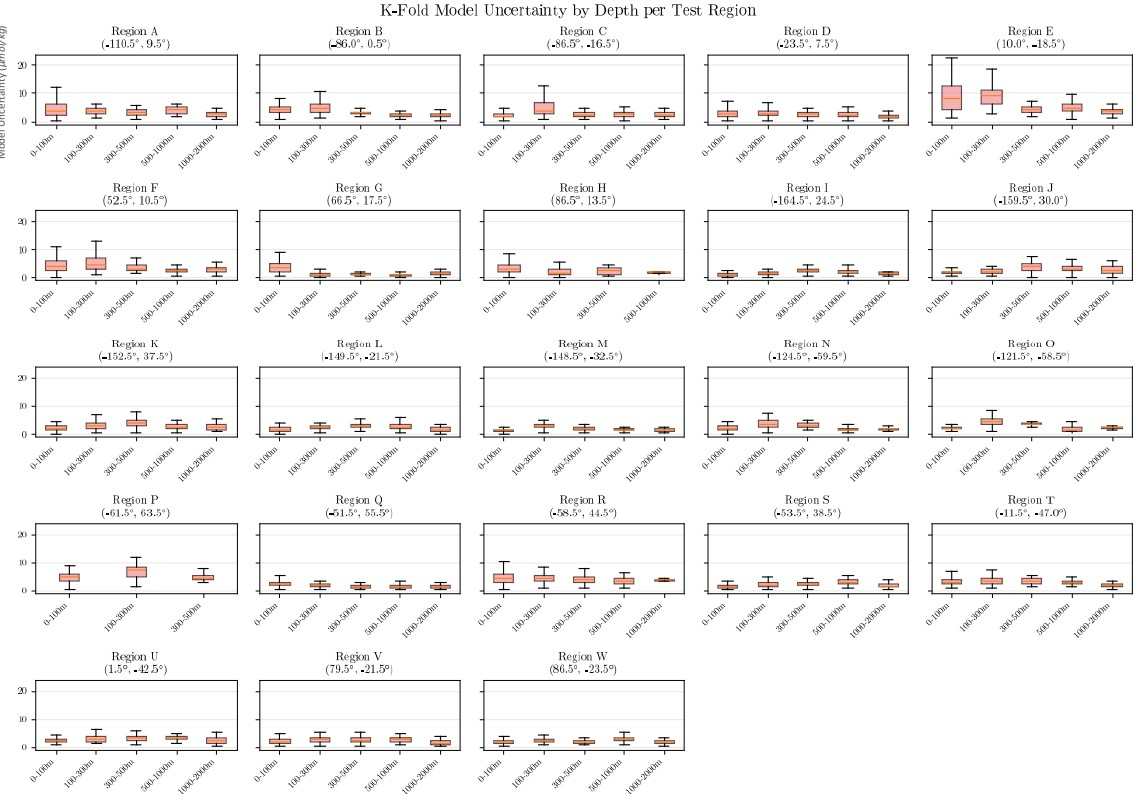

**Figure D1.** *Standard deviation of the ensemble members per depth layer.* We plot the distribution of the standard deviation for each depth layer from the surface down to $2000\ m$.

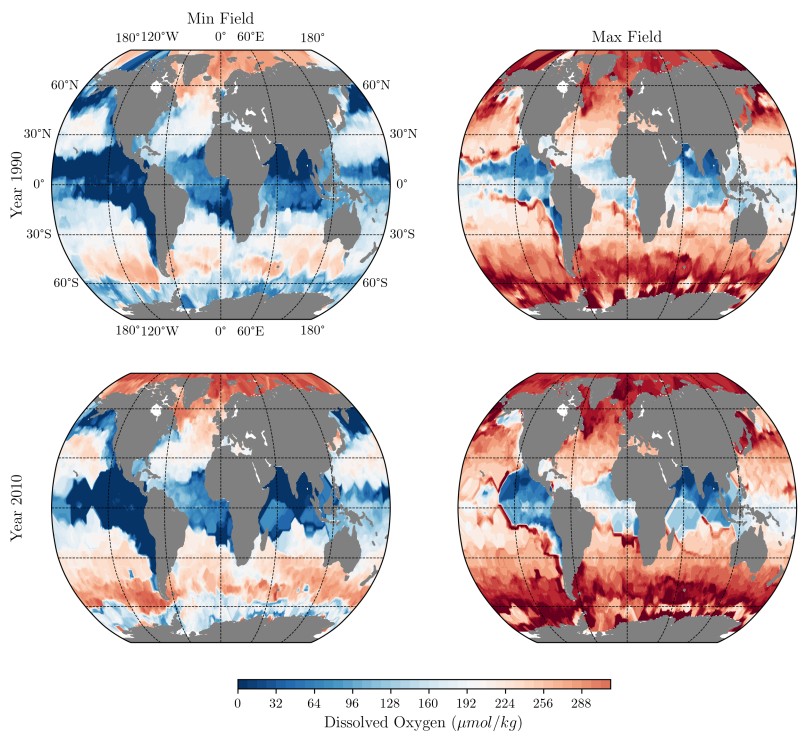

**Figure D2.** *Minimum and Maximum fields used in the QC of the emulated profiles.* First row, Minimum and Maximum fields at 200m depth, computed as described in section 2.3 for the year 1990. Second row, Same fields for the year 2010.

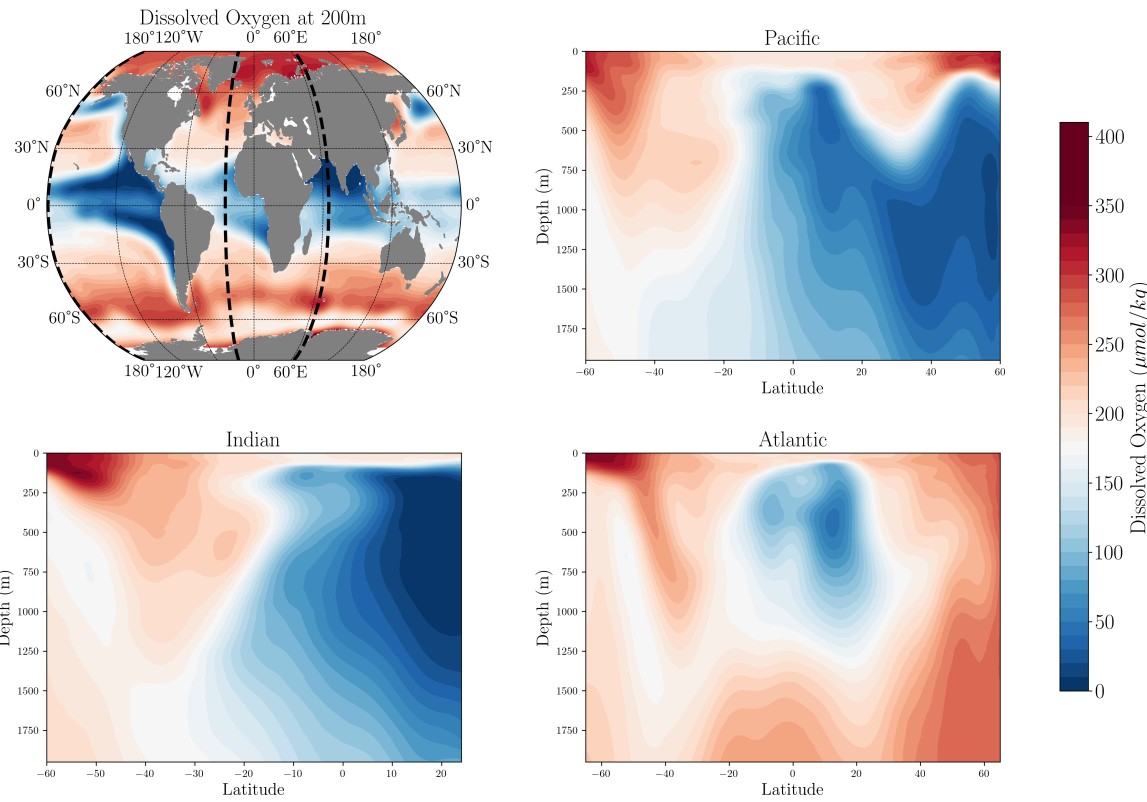

**Figure D3.** *Spatial distribution of dissolved oxygen concentration of the proposed gridded product.* The upper left figure shows the long-term mean of dissolved oxygen concentration at 200m. Dashed black lines represent the locations of the Meridional cross sections from the surface to 2000m.

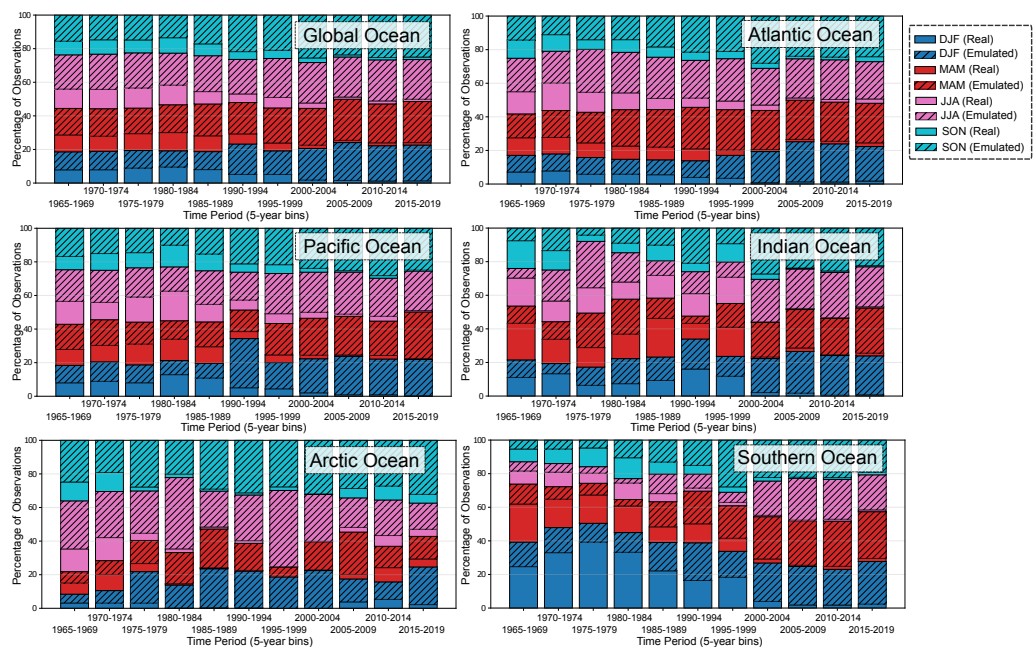

**Figure D4.** *Percentage of real and emulated data used in the interpolation by ocean region and season.* Solid bars represent real observations, while hatched bars indicate the contribution of *emulated* observations.

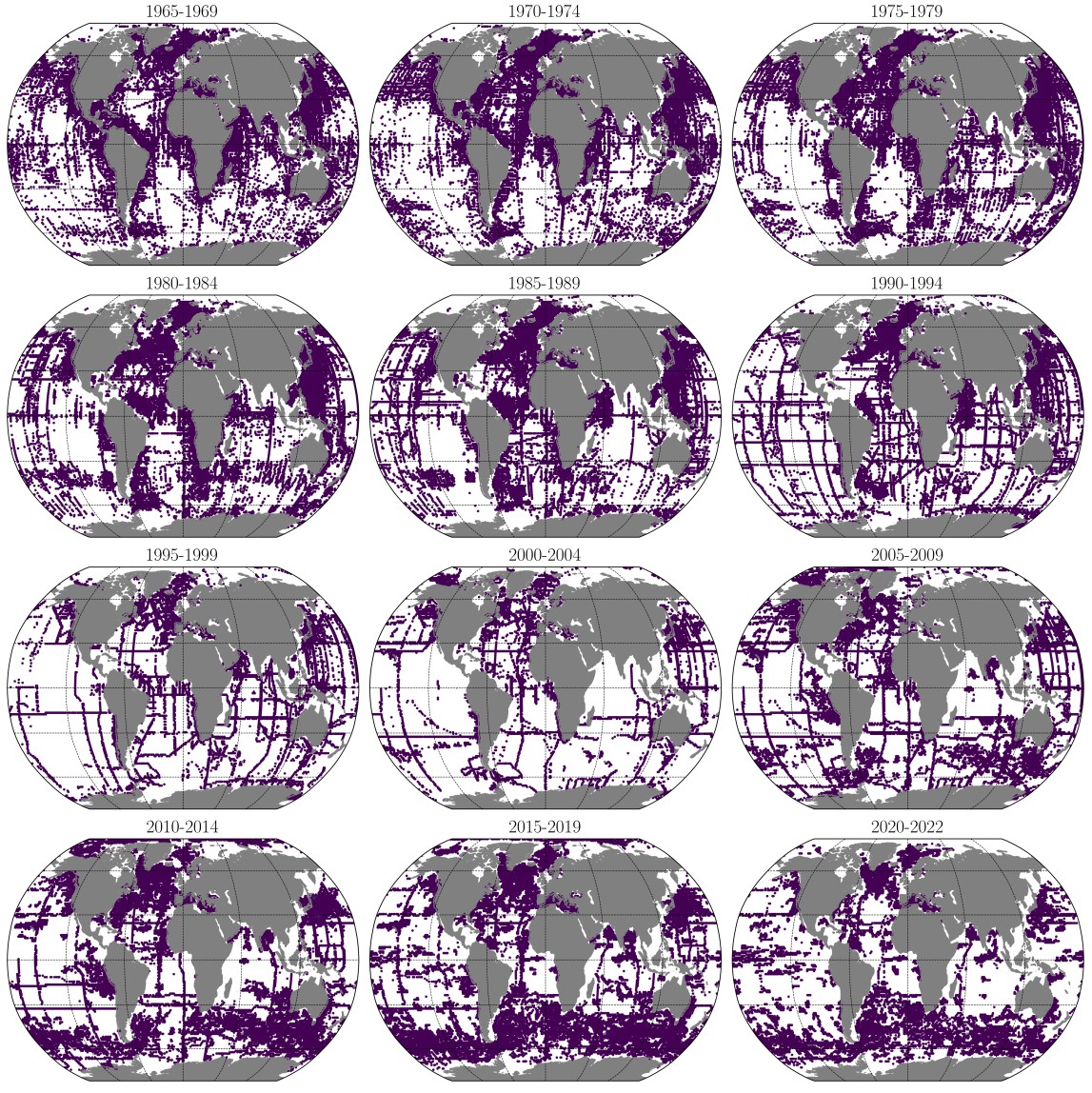

**Figure D5.** *Oxygen data coverage since 1965 at 200m depth.* Dark blue indicates $1° \times 1°$ grid locations with oxygen measurements at 5-year data intervals.

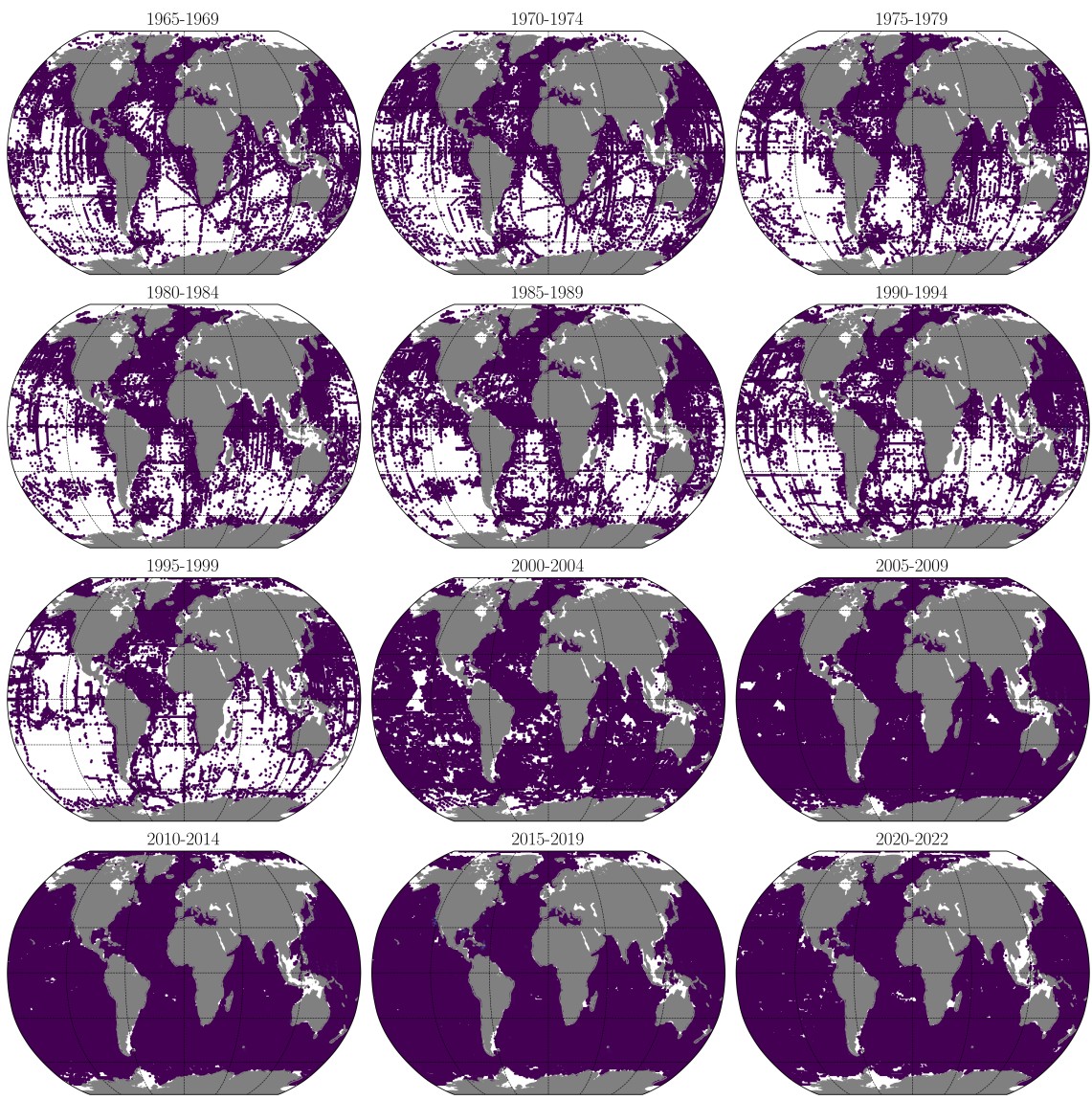

**Figure D6.** *Emulated oxygen data coverage since 1965 at 200m depth.* Dark blue indicates $1° \times 1°$ grid locations with emulated oxygen measurements at 5-year intervals. These locations essentially correspond to the coverage of temperature and salinity measurements, excluding the emulated profiles flagged as unrealistic by the QC procedure described in Section 2.3.

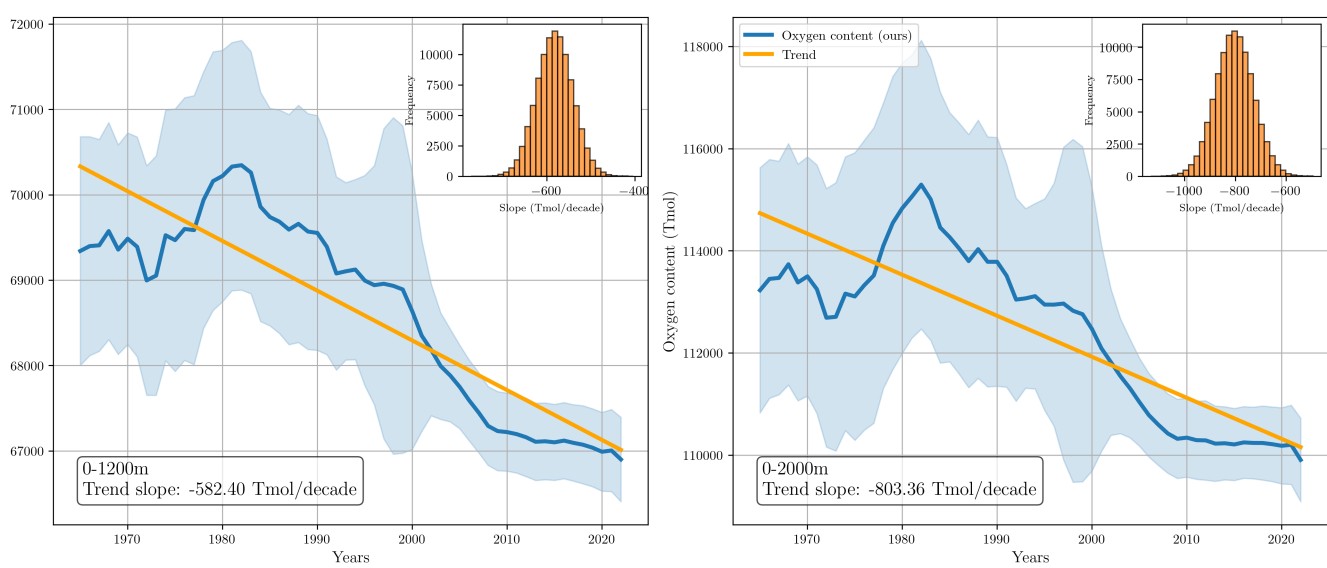

**Figure D7.** *Long-term changes in globally integrated oxygen inventory.* The left panel shows the inventory for the upper 1200 m, while the right panel displays the inventory for the upper 2000 m. The histograms in each panel represent the empirical distribution of the deoxygenation rate, computed from a Monte Carlo simulation with 10,000 realizations.