# Peer review of "A Novel Global Gridded Ocean Oxygen Product Derived from a Neural Network Emulator and In-Situ Observations"

_Earth System Science Data, 2025_

## Author Comment (AC1)

**A Novel Global Gridded Ocean Oxygen Product Derived from a Neural Network Emulator and in-situ observations**

ESSD

August 12, 2025

Said Ouala[1], Oussama Hidaoui [2], Zouhair Lachkar [3],
Contact: said.ouala@imt-atlantique.fr

[1] IMT Atlantique, CNRS UMR Lab-STICC, INRIA Team Odyssey, Brest, France.
[2] African Institute for Mathematical Sciences, South Africa.
[3] Mubadala Arabian Center for Climate and Environmental Sciences, New York University Abu Dhabi, Abu Dhabi, United Arab Emirates.

**Abstract**

The authors would like to thank the anonymous reviewer for their valuable comments and suggestions. In this document, we address the issues raised to the best of our ability. The modifications made in response to the reviewer's comments are highlighted in blue in the tracked-changes version of the manuscript.

**1 Reviewer's comments**

> **Reviewer Comment 1**
>
> I would like to thank the authors for the interesting and timely work. The paper presents a novel approach to generating a gridded dissolved oxygen product by integrating direct observations with ML-based emulations derived from temperature and salinity profiles, followed by optimal interpolation. The methodology is simple to follow and technically sound, the results are compelling, and the product demonstrates clear improvements over existing datasets, especially in capturing long-term trends and reducing uncertainties. I recommend acceptance with minor revisions, but I would like to note that my review is primarily focused on the ML aspect.

**Response**

The authors appreciate the feedback on our work. Every comment is addressed carefully below, and the modifications can be found in blue in the tracked-changes version of the manuscript.

> ### Reviewer Comment 2
>
> The training/test split was done randomly, how the authors ensure there is no data leakage? It would have been more interesting if the trains was done in a temporal way.

**Response**

We thank the reviewer for raising this important point. We agree that data leakage and the nature of the training/test split are crucial to evaluating the robustness of machine learning models.

To clarify, we used three separate subsets, training, validation, and test datasets, to design and assess our neural network model. The training and validation sets were obtained via a random split: 80% of the data was used for training and 20% for validation. This random split applies only to the training/validation phase.

The test set is completely independent of both the training and validation sets. It consists of 23 spatially distinct regions across major ocean basins, as introduced in Figures 1 and 2. These regions were excluded from the training/validation data and were used exclusively for testing, ensuring no data leakage.

We acknowledge that our original manuscript may not have made this distinction clear. To clarify, we have revised the paragraph around line 85 as follows:

"The dataset used to train the ML model consists of collocated pairs of $T$, $S$ and (DO) data from 1965 to 2022. The dataset was divided into training, validation, and test subsets. The test set comprises 23 independent $1 \times 1$ regions, distributed across all major ocean basins. The locations of these test regions, along with the performance of the machine learning-based emulator, are shown in Figure 2. The remaining data were allocated to training and validation, with 80% used for training and the remaining 20% for validation."

Regarding the reviewer's suggestion to adopt a temporal split, we chose a spatial test split instead of a temporal one for two reasons. First, we aim for the model to learn long-term deoxygenation trends related to climate change; excluding certain years from training could limit the model's ability to capture these patterns. Second, our goal was to evaluate how well the model generalizes across different oceanographic regimes, such as oxygen minimum zones or well-oxygenated regions, which motivated the use of geographically distinct test regions.

> ### Reviewer Comment 3
>
> It would have been also more robust to use a validation dataset, instead of only train/test.

**Response**

We apologize for any confusion caused by our wording. As explained in the previous response, we use separate training, validation, and test sets. The test set is fully independent from the training and validation sets, as it consists of data from locations that do not overlap

with those used for training and validation.

> **Reviewer Comment 4**
>
> Any reason why the test locations do not include any points near Europe?

**Response** We thank the reviewer for this relevant observation. There is no specific reason why the test locations do not include points near Europe. Our test regions were designed to cover all major ocean basins, including the North Equatorial and South Pacific, North and South Atlantic, and the Indian Ocean. Within each basin, we selected test locations that capture a diversity of oxygen dynamics, including both oxygen minimum zones (e.g., Regions F, G and H in the Indian Ocean, Region D in the Atlantic, and Regions A, B, C in the Pacific) and highly oxygenated regions (e.g., Regions P, Q and R). The European seas were not explicitly included because the oxygen dynamics in those areas are not different from the ones already sampled in our selected test regions.

> **Reviewer Comment 5**
>
> Any reason why using Month of the year + Day of the month in the MLP inputs instead of just using Day of the year?

**Response**

We thank the reviewer for this insightful comment. Our initial choice to use the month of the year as an input feature follows common practice in the literature (e.g., [1]), where it is shown to help capture seasonal patterns in MLP-based models. We added the day of the month to allow for finer resolution of intra-month variability, which led to a slight improvement in performance. However, our explainability analysis (XAI) confirms that this variable has relatively low importance compared to other predictors. We acknowledge that using the day of the year is a valid alternative and appreciate the suggestion.

> **Reviewer Comment 6**
>
> Can the authors describe the hyper parameter search procedure to tune the MLP?

**Response** We thank the reviewer for this important question. Our approach to tuning the MLP architecture was as follows. We began by testing the model on simulated data (a coupled ROMS and BGC model in the Indian Ocean) to assess whether an ML model could effectively predict oxygen concentrations from temperature, salinity, spatiotemporal coordinates, and surface chlorophyll-a. Initial experiments indicated that temperature and salinity alone were sufficient to achieve strong predictive performance.

We then transitioned to real data, using MODIS satellite-derived CHLA-II as an additional input. Similarly to the experiment on simulated data, we observed that including satellite CHLA-II did not improve the model's performance, so we opted to keep the architecture simple and relied only on in-situ observations to design the model.

Regarding the architecture, we performed a stepwise increase in complexity: starting with 2 hidden layers, we incrementally added more layers (up to 4) and increased the number of

[Figure]

Figure 1: Measured profiles and emulator prediction in test region J.

neurons per layer. We selected the final architecture based on the point at which additional complexity no longer yielded performance improvements on the validation set.

**Reviewer Comment 7**

Figure 1 would have been more informative if the plots where done per test region.

**Response** We thank the reviewer for raising this point. We agree with the reviewer that a per-region breakdown provides more insight. We have updated Figure 1 accordingly to show the scatter plots for each individual test region. This revised figure shows that the model performs consistently across different test regions.

**Reviewer Comment 8**

Any explanation of what's happening at depth 500 in test region J (Figure 2)?

**Response** This is a very interesting point raised by the reviewer. We analyzed the profiles at depth 500 in test region J and found that several **measured** profiles in this region show abrupt variations around 500 m depth, leading to an inflated standard deviation at that depth level. These anomalies are likely due to sensor errors rather than physical processes. Figure 1 illustrates these outliers by comparing measured and predicted profiles. Notably, the emulator outputs remain smooth and do not reproduce these irregular patterns.

**Reviewer Comment 9**

It would be interesting to use any XAI method to study feature importance for the MLP.

**Response**

We appreciate the comment from the reviewer. We have added a new section in the appendix presenting an analysis of feature importance using Integrated Gradients. The results confirm that the most influential features for predicting dissolved oxygen are the geographical location of the profiles (latitude and longitude) along with physical variables, namely temperature and salinity. These features reflect the importance of the regional context through dominant physical ventilation regimes, biogeochemical dynamics, and oxygen solubility in explaining oxygen variability in the ocean.

> **Reviewer Comment 10**
>
> Any plans to share the code used and not only the dataset?

**Response**

We thank the reviewer for raising this point. We do plan to share the code. Currently, the code consists of several modules developed and hosted by different contributors, covering data extraction and preprocessing, model training, emulation, quality control of emulated profiles, and interpolation. Due to this "distributed" development, publicly releasing the code in its current form is not feasible. However, we are happy to provide it upon request.

However, we are in the process of cleaning and organizing the code to make it publicly available as a single package. We appreciate the reviewer's interest and are committed to ensuring the code is shared as soon as it reaches an appropriate level of clarity and documentation.

> **Reviewer Comment 11**
>
> Typos:
> * Line 35: "weather forecasting" instead of "forecasting"
> * Many citations are badly formatted, /citet vs /citep

**Response** We thank the reviewer for spotting these typos. They have been corrected in the new version of the manuscript. Regarding the citations, all references are cited in the text using the /citet.

**References**

[1] Takamitsu Ito, Ahron Cervania, Kaylin Cross, Sanika Ainchwar, and Sara Delawalla, "Mapping dissolved oxygen concentrations by combining shipboard and argo observations using machine learning algorithms," *Journal of Geophysical Research: Machine Learning and Computation*, vol. 1, no. 3, pp. e2024JH000272, 2024.

---

## Author Comment (AC3)

**A Novel Global Gridded Ocean Oxygen Product Derived from a Neural Network Emulator and in-situ observations**

**ESSD**

October 22, 2025

Said Ouala1, Oussama Hidaoui 2, Zouhair Lachkar 3, Contact: \myEmail

 $^1$  IMT Atlantique, CNRS UMR Lab-STICC, INRIA Team Odyssey, Brest, France.  $^2$  African Institute for Mathematical Sciences, South Africa.  $^3$  Mubadala Arabian Center for Climate and Environmental Sciences, New York University Abu Dhabi, Abu Dhabi, United Arab Emirates.

**Abstract**

The authors would like to thank the anonymous reviewer for their valuable comments and suggestions. In this document, we address the issues raised to the best of our ability. The modifications made in response to the reviewer's comments are highlighted in red in the tracked-changes version of the manuscript. The modifications in green relate to comments that were raised by both reviewers. Please note that, as we have revised the manuscript, the numbering of figures and sections referenced in our answers refers to the revised version and may differ from that of the original submission.

**1 Reviewer's comments**

**Reviewer Comment 1**

I thank the authors for this interesting work. I think adding the emulated data in combination with Optimal Interpolation is a useful approach of further developing and improving observation based 4D reconstructions of oxygen. Making these reconstructions and using them to identify and explain variability on different scales is a very important goal and the link with PDO looks promising.

**Response**

The authors appreciate the feedback on our work. Every comment is addressed carefully below, and the modifications can be found in red in the tracked-changes version of the manuscript. Comments in green refer to modifications suggested by both reviewers.

The Optimal Interpolation method seems sound, even though the manuscript would benefit from additional details on why these parameters were chosen, especially regarding the differences of the two separate products.

**Response** We thank the reviewer for this comment. The choice of OI parameters is mainly based on the data availability in the periods before and after the ARGO era. Specifically:

- The yearly product uses a 5-year aggregation window and a 1000 km correlation length scale due to the relatively sparse historical record. These parameters are based on the work of Ito [2022] who uses these values in the same period.
- The monthly product, which covers the ARGO era is based on a 3-month aggregation and 300 km correlation length scale. These parameters are based on temperature and salinity interpolated ARGO product [Gaillard et al., 2016].

Both products are based on a 1-degree grid, which is the standard choice for most of the objectively analyzed products that are based on in-situ data [Cheng et al., 2017, Ito, 2022], including the ML-based products [Sharp et al., 2022, Ito et al., 2024a]. The vertical range was limited to 0-2000 m since 2000m is the maximum depth of most of the ARGO data profiles [Roemmich et al., 2009] and it corresponds to the extent that covers most of the relevant oxygen variation phenomena, including oxygen minimum zones.

We agree with the reviewer regarding the lack of discussion on these parameters. We added the following paragraph to justify the choice of these parameters in the OI section:

"We perform an interpolation of both observed and emulated dissolved oxygen data using a standard Optimal Interpolation (OI) method. Following the substantial increase in temperature and salinity data coverage in the ARGO era (after 2002), which also corresponds to an increase in emulated oxygen profiles, we construct two gridded products with different temporal resolutions. The first product, with yearly resolution from 1965 to 2022, is designed to study decadal and climate-change-related variability. The second product, with monthly resolution, focuses on the ARGO era (2003-2022) to capture seasonal and interannual variability. Both products use a 1° horizontal resolution, which is standard among most objectively analyzed products based on in-situ data [Cheng et al., 2017, Ito, 2022], including ML-based reconstructions [Sharp et al., 2022, Ito et al., 2024a]. The vertical grid comprises 65 standard WOD depth levels ranging from 0 m to 2000 m, matching the typical vertical extent of most ARGO flaots [Roemmich et al., 2009]. "

We also further motivate the choice of the e-foolding length scale Lref in lines 190:

"where  $\sigma_b^2$  is the total variance of the background field, and  $L_{m,n}$  is the distance between two grid points m and n.  $L_h$  is the e-folding horizontal length scale. In this work, we follow the approach of Ito [2022] and set  $L_h$  to 1000 km for the interpolation of the yearly product between 1965 and 2022. For the monthly product, a larger number of observations are available, allowing us to reduce  $L_h$  to 300 km, which is consistent with ARGO-based products of temperature and salinity fields [Gaillard et al., 2016]. "

The machine learning part and therefore the quality of the emulated data is not tested robustly enough. There is only the comparison with "test regions" taken from the validation dataset. There needs to be more detail on how these were chosen and how independent they actually are. The world map in figure 8 shows a good coverage of ocean data, but in figure C3 and C4 you still have gaps and have not analysed any seasonal bias. The validation data could also have been partly seen by the machine, since it is often used to control learning and prevent overfitting. There are also no other measures of machine learning performance. It would've been good to have an n-fold machine learning ensemble - is there a large spread in the predictions? Do values for some areas differ substantially from one ensemble member to another?

**Response**

We thank the reviewer for raising this important comment. Please find below our detailed responses to each of the points raised:

• Regarding the comparison with "test regions" taken from the validation dataset: To clarify, we used three separate subsets, training, validation, and test datasets, to design and assess our neural network model. The training and validation sets were obtained via a random split: 80% of the data was used for training and 20% for validation. This random split applies only to the training/validation phase.

The test set is completely independent and is neither seen nor partly seen by the training or validation sets. It consists of 23 spatially distinct regions across major ocean basins, as introduced in Figures 1 and 2. These regions were excluded from the training/validation data and were used exclusively for testing, ensuring no data leakage.

We acknowledge that our original manuscript may not have made this distinction clear. To clarify, we have revised the paragraph around line 90 as follows:

"The dataset used to train the ML model consists of collocated pairs of *T*, *S* and (DO) data from 1965 to 2022. The dataset was divided into training, validation, and test subsets."

• Regarding the choice of the test regions: Our test regions were designed to cover all major ocean basins, including the North Equatorial and South Pacific, North and South Atlantic, and the Indian Ocean. Within each basin, we selected test locations that capture a diversity of oxygen dynamics, including both oxygen minimum zones (e.g., Regions F, G and H in the Indian Ocean, Region D in the Atlantic, and Regions A, B, C in the Pacific) and highly oxygenated regions (e.g., Regions P, Q and R). To emphasize this point, we clarified in the paper why we choose these locations as follows:

"The dataset was divided into training, validation, and test subsets. The test set comprises 23 independent  $1^{\circ} \times 1^{\circ}$  regions, distributed across all major ocean basins, including the North Equatorial and South Pacific, North and South Atlantic, and the Indian Ocean. Within each basin, we selected test locations that capture a diversity of oxygen dynamics, including both oxygen minimum zones (e.g., Regions F, G, and H in the Indian Ocean, Region D in the Atlantic, and Regions A, B, and C in the Pacific) and

highly oxygenated regions (e.g., Regions P, Q, and R). The locations of these test regions, along with the performance of the machine learning-based emulator, are shown in Figure 3. The remaining data were allocated to training and validation, with 80% used for training and the remaining 20% for validation. "

• Regarding the seasonal bias: We thank the reviewer for this comment and we agree that it's an important aspect of in-situ data, particularly the data based on cruises. We added a figure (D3) in the appendix D that analyses the seasonal bias in the sampling of the real and emulated oxygen data. Overall, we observe the presence of a seasonal bias, especially at high latitudes and in the southern ocean. However, this bias is substantially reduced during the Argo era. This analysis further supports our idea of using emulated profiles derived from Argo temperature and salinity data, which significantly improve the data coverage of oxygen data and reduce seasonal bias, particularly in the ARGO era.

We have also made the caption of Figure 9 more precise as the panel a of this figure represents the total number of real/emulated DO2 data that was gridded in the 1° grid.

- on the fact that validation data could also have been partly seen by the machine, since it is often used to control learning and prevent overfitting: We recall here that, as discussed above, the test set is completely independent and is not used in the training process of the model.
- There are also no other measures of machine learning performance. It would've been good to have an n-fold machine learning ensemble - is there a large spread in the predictions? Do values for some areas differ substantially from one ensemble member to another? We agree with the reviewer regarding the measure of the uncertainty of the ML model. Initially, we actually built an ensemble of ML models based on an n-fold training where we aimed at using the discrepancy of the models as a measure of the uncertainty of the emulated profiles. Overall, the spread within the machine learning (ML) ensemble is relatively small, with no specific regions exhibiting notably high standard deviations. For instance, Figure 1 shows the distribution of the standard deviation of the n-fold ML ensemble predictions across water layers from the surface down to 2000 m. The ensemble standard deviations are generally low, with median uncertainties within each depth being around 5  $\mu$ mol kg-1. This emphasizes that the training of the ML model is stable and that the n-fold training methodology is able to recover some of the epistemic uncertainty [Valdenegro-Toro and Mori, 2022] related to limitations in the data coverage and/or model parameterization. Importantly, we do not use these uncertainties as measurement errors in the optimal interpolation (OI) as it leads to the interpolation being overconfident in the emulated profiles. Specifically, these n-fold estimates of the model uncertainty are missing the aleatoric uncertainty that is inherent to the training data-itself, resulting in standard deviations that are sometimes smaller than the measurements errors of the true oxygen data (typically on the order of  $\sim 3\%$  of the oxygen value) used in the OI. Instead, we assign an uncertainty equal to 4% of the emulated oxygen value when performing OI, which ensures that more weight is given towards real observations.

We added the following section on the main paper to discuss the uncertainty of the MLP model:

To evaluate the epistemic uncertainty [Valdenegro-Toro and Mori, 2022] of training a machine learning model to predict ocean oxygen concentration, we start by training a k-fold ensemble of MLP models (with k = 5), each one based on different training and validation datasets. The trained models are then used to predict oxygen data in the test set. Figure 1 shows the distribution of the standard deviation of the k-fold ML ensemble predictions across water layers from the surface down to 2000 m. Overall, the ensemble standard deviations are generally low, with median uncertainties within each depth layer around 5  $\mu$ mol kg-1. This emphasizes that the training of the ML model is stable and that the k-fold training methodology is able to recover some of the epistemic uncertainty [Valdenegro-Toro and Mori, 2022] related to limitations in the data coverage and/or model parameterization. However, using this uncertainty as the error estimate for the emulated profiles in the optimal interpolation would have made the interpolation overconfident in the emulated profiles relative to the real observations. Therefore, we use only a single MLP model (the best-performing one) for emulating the profiles used in the interpolation. This model is further evaluated in the subsequent analyses. The associated error estimate of this model, described in Section 2.4.3, is set higher than that of the real observations.

**Reviewer Comment 4**

I am also not fully convinced that the decadal and synoptic variability you've seen or any additional features you observed is definitely real. That said, I do not exclude the possibility that it is real. I think it is important to explain further why you think it is real - because that's the main issue faced by anyone using interpolation techniques. Currently, it is not clear to the reader why these results couldn't still be a product of the sparsity of ocean observations. After all, even if you added many datapoints based and temperature and salinity, this new dataset is still sparse given how vast the ocean is. In some parts of the text it sounds like by simply observing decadal variability it can be declared an improvement. It could indeed represent an improvement, but the manuscript needs to provide stronger justification. One way of doing this could be a model validation, for example. You could also look at which basins are driving this decadal variation and discuss the processes in these basins that could drive the decadal variability. You do this partly in the text with PDO, but the manuscript could benefit from a more detailed analysis.

**Response**

We thank the reviewer for raising this important point. We agree with the reviewer, and we think it is important to provide some evidence on why we think the decadal or synoptic variability we see in our product are real and are not the result of some interpolation bias.

Since we are using optimal interpolation, the two main interpolation biases we can have that can significantly influence both the qualitative and statistical variability of the resulting product are due to i) regions with a significantly small number of data or ii) artifacts due to bad data. We explain below why think that these biases, while being present in our interpolation due to measurements errors and lack of data in some regions despite the emulated profiles, are not likely to be the drivers of the decadal and synoptic variability we see.

• On the realism of the decadal variability: While a detailed analysis of the role of different modes of decadal and interannual variability on oxygen content is beyond the scope of the present study—which we would like to keep focused on presenting the O2 reconstruction product, its methodology, and its comparison with previous ML-based and observation-only reconstructions—this topic will be the subject of a separate study that we are currently preparing, dedicated to interannual and decadal oxygen variability. Nevertheless, the decadal variability highlighted in the present paper is consistent with previous modeling studies that reported significant oxygen variability on decadal timescales [Oschlies et al., 2018, Deutsch et al., 2011]. Moreover, the variability in the deoxygenation rate revealed by our product agrees with recent studies linking decadal modulation of deoxygenation in the tropical Pacific Ocean to changes in PDO phases [Duteil et al., 2018, Poupon et al., 2023]. We do not believe this variability arises from interpolation biases due to limited data availability, as a lack of data in an optimal interpolation method typically results in a gridded field with near-zero anomalies (i.e., a field close to climatology). This type of bias was studied in the context of estimating ocean deoxygenation [Ito et al., 2024b], and its impact is known to result in an underestimation of ocean deoxygenation. Therefore, such biases would have tended to produce weaker deoxygenation rates during the pre-ARGO period of limited observations compared to the ARGO period, when observation density increased nearly tenfold. However, in our product, the deoxygenation rate is actually weaker during the ARGO period (2003–2022) than in the pre-ARGO period (1980–2000).

Regarding a bias due to data with bad quality, as we discussed in the paper, we actually observe this correlation between deoxygenation and the PDO in numerical models, which makes the possibility of an artificial correlation in our product unlikely.

To emphasize these points, we have 1) strengthened the discussion of the link between decadal variability in oxygen content and major climate variability modes, such as the PDO, by adding references to three additional studies that have explored this relationship using model simulations, and 2) explicitly clarified that the variations in the rate of deoxygenation in our product are unlikely to result from interpolation biases associated with the scarcity of observations. The revised discussion now reads:

Finally, it is worth noting that our product reveals a much stronger decadal and interdecadal variability in the rate of deoxygenation compared to previous ML-based reconstructions (Figure 8). The influence of decadal climate variability on regional and global deoxygenation is well established [Oschlies et al., 2018]. For instance, the rate of deoxygenation in our product from 1980 to the early 2000s was substantially higher than during the 1960s and 1970s, as well as over the past decade. These variations, largely absent in earlier ML-based reconstructions, are consistent with model-based studies suggesting that major climate variability modes, such as the Pacific Decadal Oscillation (PDO), strongly influence ocean oxygen content [Deutsch et al., 2011, Duteil et al., 2018, Ito et al., 2019]. For example, Deutsch et al. [2011] showed that PDO explains about 24% of the variability in the volume of suboxic waters in the Pacific based on a model simulation, attributing this relationship to PDO-driven modulation of trade winds, thermocline depth, and respiration rates in the eastern tropical Pacific. Duteil et al. [2018] demonstrated that the sluggish equatorial circulation during positive PDO phases (such as in the 1980s and 1990s) results in a pronounced deoxygenation in the

eastern equatorial Pacific and an intensification of its OMZ. Ito et al. [2019] further emphasized the importance of PDO-driven vertical displacements of isopycnals in modulating tropical Pacific ocean oxygen content. More recently, Poupon et al. [2023] showed that deoxygenation is favored during positive phases of the PDO, whereas negative PDO phases, dominant in the 1960s, 1970s, and over the past two decades, enhance oxygenation in the tropical Pacific, thereby partly offsetting anthropogenic or climate change–driven deoxygenation. Importantly, these decadal oxygen variations are unlikely to arise from interpolation biases related to data scarcity. According to previous work by Ito et al. [2024b], such biases would have tended to produce weaker deoxygenation rates during the pre-ARGO period of limited observations compared to the ARGO period, when observation density increased nearly tenfold. However, in our product, the deoxygenation rate is actually weaker during the ARGO period (2003–2022) than in the pre-ARGO period (1980–2000).

• On the realism of the synoptic variability: The synoptic variability highlighted in Fig. 6 cannot be attributed to a lack of data, as such an interpolation bias would have instead resulted in near-zero anomalies. In contrast, we observe eddy-like coherent oxygen structures with both positive and negative anomalies, indicating that observations were effectively used to reconstruct these features.

Regarding the quality of the data used to retrieve these structures, we also consider it unlikely that the observed synoptic variability arises from erroneous profiles. The variability patterns are remarkably consistent across the three compared products, the main difference, as shown in Fig. 6, is that the synoptic-scale eddies appear more energetic in our reconstruction. A visual inspection of the other ML-based products further confirms that similar structures are present, making the hypothesis of artifacts from poor-quality data highly improbable.

We motivated the realism of the synoptic scale variability in the paper as follows:

Beyond the climatological spatial distribution, we assess the spatial resolution of the proposed monthly DO product relative to state-of-the-art machine learning-based gridded datasets. Figure 6 presents an example of the DO anomaly field in the equatorial Pacific ( $-179^{\circ}$  E to  $-100^{\circ}$  E,  $30^{\circ}$  S to  $30^{\circ}$  N) alongside the vertically and monthly averaged Radially Averaged Power Spectral Density (RAPSD) of our product, compared with GOBAI-O2 and Ito et al. (2024). This region is characterized by energetic synoptic-scale variability [Chelton et al., 2007], and we evaluate whether our product better captures these processes. The RAPSD indicates 100-200% higher energy levels at wavelengths around  $O(10^3)$  km, compared to the ML baselines, suggesting an improved representation of small-scale variability. This is further illustrated through the visual analysis of the anomaly field in Figure 6, where our product better represents finer synoptic-scale structures on the order of  $O(10^3)$  km, revealing more energetic mesoscale eddies than the other ML-based products. These variations are unlikely to result from interpolation biases associated with data scarcity, which would instead tend to produce near-zero anomalies. Likewise, the possibility that they arise from spurious or low-quality profiles is also unlikely, as similar patterns are consistently observed across the other ML-based products.

The citing commands need to be checked, especially the difference between cite and citeA if you're writing in Latex. Author et al. (2025) vs (Author et al., 2025).

**Response** We thank the reviewer for spotting the problem with our citations. They have been corrected in the new version of the manuscript. All references are cited using /cite when they are discussed in the text and using /citep otherwise.

**Reviewer Comment 6**

L27: You should also mention the seasonal bias. Especially for Polar regions there are still few datapoints in the winter. There is also different data availability in different decades - and different quality of data.

Response We thank the reviewer for this comment. We added the following text:

Despite advances in autonomous profiling floats, underwater vehicles, and large-scale ocean sensing programs such as ARGO, dissolved oxygen observations remain insufficient to accurately estimate deoxygenation rates at both global and regional scales [Gruber et al., 2010, Claustre et al., 2020]. Large regions, particularly during the pre-ARGO era, in the South Pacific, the Indian Ocean, and the polar regions remain undersampled [Hermes et al., 2019, Grégoire et al., 2021] and the presence of seasonal biases and irregular sampling, especially before the ARGO era, significantly limits the ability to directly analyze fine-scale spatio-temporal variability from observations.

**Reviewer Comment 7**

L39: They are not just a reference for model calibration, they provide important observation based estimates of the oxygen budget.

**Response**

We agree with the reviewer on this point and modified the paragraph which now mentions oxygen budget as well.

In this context, assessing global and regional changes in ocean oxygen content and budget requires developing interpolation methods that map available data onto a regular space-time grid. Gridded oxygen products also play a crucial role in validating ocean models, including both global models used in Earth System Models (ESMs) and regional models necessary for projecting the impact of climate change on oxygen at regional scales.

**Reviewer Comment 8**

L48: Regarding marginal seas not present in other studies: That's true, but your validation dataset in figure 2 looks like you are also not focusing on them.

**Response** We thank the reviewer for this comment. It is true that our test regions do not include marginal seas. However, our statement in that section referred to the comparison

between gridded products, not to the ML-based emulation specifically within marginal seas. A rigorous analysis of the interpolated fields in marginal seas would require a dedicated regional study, which we consider beyond the scope of the present paper, as it would considerably extend its length. Instead, we focused on evaluating the global patterns and variability of ocean oxygen and on comparing our results with existing baseline products.

**Reviewer Comment 9**

L48: Good point, I think it is important to address that.

**Response**

We thank the reviewer for their positive comment. While state-of-the-art ML-based mappings of dissolved oxygen typically rely on interpolated products to generate gridded fields, our approach operates at the profile level. Specifically, our ML model predicts individual profiles, which are then jointly interpolated with real observations using Optimal Interpolation (OI). This framework provides better control over the interpolation process, particularly by allowing us to assign lower weights to the emulated data compared to real observations. In practice, this is achieved in OI by prescribing larger measurement uncertainties for the emulated profiles than for the actual observations.

**Reviewer Comment 10**

L50: Perhaps this will be a subject later - but why did you choose to start from 1965? Isn't that optimistic given we have very few datapoints during that time? What about data/measurement quality?

**Response** We chose to start from 1965 since, as shown in figure D3 in the appendix, there are still a good amount of points in this period that make possible starting the interpolation from 1965. It was also considered in earlier state-of-the-art studies [Ito, 2022]. We added the following footnote to explain our choice:

We chose to start from 1965, as this year was used in previous state-of-the-art reconstructions [Ito, 2022], and it corresponds to a period with sufficient data coverage, as shown in Figure D4 in the Appendix.

**Reviewer Comment 11**

L62: How do you test that your product really fares better in regions where there is no data?

**Response** Our comment around L62 was referring to Figure 8, where we show that the gridded product with emulated profiles significantly reduces the uncertainty with respect to optimal interpolation of only dissolved oxygen data. This is mainly due to having a better sampling of the observations due to adding the emulated profiles.

L86: Do you only train once? Or do you use a machine learning ensemble, where the 20% are different for each ensemble member, meaning that eventually you've every datapoint at least once in training before calculating an ensemble mean.

**Response** We thank the reviewer for this comment. The results we show are based on a single MLP model trained once on a specific train/validation split. We actually also conducted a n-fold ML ensemble (with n=5), but we found that the spread in the ML predictions is small and would lead to an OI that is more confident in the emulated profiles than the real profiles. For this reason we focused our analysis on a single MLP as its the one used in the interpolation.

However, and as discussed in our answer to comment 3, we added a figure that evaluates the uncertainty of the MLP model computed from a n-fold MLP ensemble.

**Reviewer Comment 13**

L88: How did you choose the test regions? Did you use an algorithm like the SOM method by Landschützer et al. 2016?

**Response** We thank the reviewer for this comment. We already answered this comment in our answer to comment 3.

**Reviewer Comment 14**

L95: Do you mean Multilayer Perceptron? Perhaps you should also mention that this is a feedforward neural network, which may be more familiar for many readers. Also, there are many different architectures of neural networks.

**Response** We agree with the reviewer on this comment, we added the reference to a feedforward NN in the text as follows:

The machine learning model used to emulate DO data is a Multilayer Perception (MLP) model (also referred to as a feedforward or fully connected neural network).

**Reviewer Comment 15**

L101: Perhaps it would be good to give a general summary in the text. You don't need to provide numbers of layers here but at least tell the reader how hyperparameters were chosen. It would be good to provide detail on why you chose this architecture.

**Response**

We thank the reviewer for this comment. We added the following sentence in the appendix to explain why we came up with this architecture. The MLP architecture and hyperparameters were selected through incremental testing, starting from a simple configuration and gradually increasing model complexity until no further improvement was observed on the training of the model.

L115: Regarding errors in earlier years: this is actually one of the main issues we face - perhaps a few words more need to be added in the introduction instead of doing this here (in addition to the sparse regions and seasonal bias I mentioned).

Response We agree with the reviewer on this comment and we added the following text in the introduction: Large regions, particularly during the pre-ARGO period, in the South Pacific, the Indian Ocean, and the polar regions remain undersampled [Hermes et al., 2019, Grégoire et al., 2021] and the presence of seasonal biases and irregular sampling, especially before the ARGO era, significantly limits the ability to directly analyze fine-scale spatiotemporal variability from observations.

**Reviewer Comment 17**

Figure 1: This looks like a good match between predicted and true values, but the scatter markers mask each other when it gets more crowded. It would be clearer to make a density plot (where colours indicate the number of datapoints at that location in the plot), similar to figure 3. That way you can also see where most of the datapoints are and where the outliers are.

**Response** We agree with the reviewer regarding the density of the observations in the scatterplot. We modified this figure to highlight the model fit per test region. This revised figure is visually less crowded since we use different colors for each region and shows that the model performs consistently across different test regions.

**Reviewer Comment 18**

L124-126: Perhaps I'm misunderstanding something, but it is not clear how you deal with data in sparse regions (i.e. regions where you don't have any or much historical data). I know this is not easy to do, but it is important to address.

**Response** We actually explain in the following paragraphs how we deal with regions with a small amount of data. For each  $1^{\circ}$  grid cell, the min-max values are computed based on a neighborhood of grid cells. The size of the neighborhood of cells increases until a sufficient number of 15 observations is collected.

**Reviewer Comment 19**

L139: 0 m to 2000 m: It would be good to say why you chose these limits.

**Response** We thank the reviewer for this comment, we addressed this comment in our response to the reviewer comment 2.

Figure 3: Instead of longitude and latitude I think it would be more informative to use ocean basins. Depth and Year look good - although I would also be interested in data shallower than 300 m. Minor point: perhaps reverse the y-axis here, so that deeper levels are down, like in the real ocean.

**Response** We agree with the reviewer and thank them for this comment. We revised Fig. 3 as suggested by the reviewer as follows:

- We used the identifiers of the test regions instead of longitude-latitude
- We refined the depth boxplot analysis in the upper layers.
- We inverted the y-axis of the depth boxplot.

**Reviewer Comment 21**

L156: It wasn't clear before that you planned to do two separate products: one yearly product and one monthly product each with different time ranges. Why did you chose these years and parameters?

**Response**

We agree with the reviewer that the motivation behind having two separate products is lacking in the text. We motivated this choice in our response to the reviewer's comment 2.

**Reviewer Comment 22**

L166: Regarding emulation uncertainty: On other machine learning work this is done via the standard deviation of the machine learning ensemble (e.g. MOBO-DIC in Keppler et al. 2023). Perhaps I'm misunderstanding this, but I'm not sure how you justify the slight increase to 4%. Perhaps it would be good to make that clearer.

**Response** We thank the reviewer for this comment. We addressed this comment in our answers to the reviewer comments 3 and 12.

**Reviewer Comment 23**

L194: You say your product better resolves synoptic scale structures. Why could this be? And how confident are you that this is real? Also, if you use "resolve" it sounds like you could be referring to resolution, but both GOBAI-O2 and Ito et al. 2024 use the same resolution of 1 degree.

**Response** We thank the reviewer for this comment. We already addressed the reviewer's concern on the realism of the synoptic scale variability in our response to the comment 4.

When we use the term "better resolve" we are referring to the fact that the spectral analysis

of our product exhibits higher energy levels at these scales compared to other ML-based products. These higher energy levels reflect that our product better represents (or resolves) these structures.

We agree with the reviewer on the fact that using the term "resolve" here can lead to confusion, so we replaced it with "represents" as follows:

This is further illustrated through the visual analysis of the anomaly field in Figure 6, where our product better represents finer synoptic-scale structures on the order of  $O(10^3)$  km, revealing more energetic small-scale eddies than the other ML-based products.

**Reviewer Comment 24**

Figure 5: I like that you examined the wavelength in such a way. Could you perhaps add one more wavelength label on the x-axis, so that it's clear at what wavelengths the other differences are?

**Response** We thank the reviewer for their positive comment on our figure. We updated the figure with more labels on the x-axis.

**Reviewer Comment 25**

L196: It is not clear that you are now talking about your monthly product.

**Response** We agree with the reviewer on this comment. We added in the paper explicit references to which product we are using.

Example of the climatological spatial distribution:

We first compare the climatological spatial distribution of dissolved oxygen in our product, derived from the yearly dataset because of its longer temporal coverage, to the WOA23 baseline (Figure 5).

Example of the spatial resolution analysis:

Beyond the climatological spatial distribution, we assess the spatial resolution of the proposed monthly DO product relative to state-of-the-art machine learning-based gridded datasets.

**Reviewer Comment 26**

L197: Minor point: You can just say "compare with GOBAI-O2 and Ito et al. 2024" instead of "previous ML-based products, including GOBAI-O2 and Ito et al. 2024". Otherwise, it sounds like you have more products to compare with.

**Response** We agree with the reviewer on this point, we corrected the sentence as follows:

We also compare the climatological seasonal cycle of oxygen in our product across both hemispheres with that from WOA23, as well as with the GOBAI-O2 product [Sharp et al., 2022] and the Ito et al. (2024) product [Ito et al., 2024a].

L201: Same here, fewer words might be easier on the reader. Just say GOBAI-O2 and Ito et al. 2024.

**Response**

We also agree with the reviewer here. We simplified our sentence as suggested by the reviewer

**Reviewer Comment 28**

Table 1: To a reader not familiar with this correlation it would be good to explain why this is desired and important.

**Response** We thank the reviewer for this comment. Near the surface, oxygen concentrations are close to saturation levels, the variability of which is primarily driven by temperature. Therefore, we expect the strong coupling between oxygen and temperature seasonality to be accurately captured in the oxygen reconstruction product.

We added the following text to the revised paper to explain why we are looking into the correlation coefficient.

As expected, Figure 7 shows that oxygen seasonality is more pronounced in the upper ocean, reflecting the strong seasonal variability in oxygen saturation, which is primarly driven by temperature variations. This relationship is further quantified by the Pearson correlation coefficients between dissolved oxygen and temperature anomalies reported in Table 1.

**Reviewer Comment 29**

L209: Is there any ocean further south than -80 degrees?

**Response** We thank the reviewer for his comment. In our comparison to the ML based baselines, we excluded the polar regions in the Arctic (>80°N) and Southern (<80°S) oceans since they were not included in GOBAI-O2. We agree that most of the Southern Ocean lies north of 80°S.

**Reviewer Comment 30**

L217: Minor point: there are a varying number of blank spaces around the plus-minus signs.

**Response** We thank the reviewer for noticing this. It is not a typo on our side, as all the plus-minus signs are written consistently in math mode. The variation in spacing is likely due to LaTeX rendering.

L219: The reference to the "full 1965-2022 period" again highlights that the manuscript does not clearly distinguish between the different uses of the yearly and monthly products across their respective time ranges. In particular, the monthly product covering 2005-2022 has so far been rarely discussed, and its role relative to the yearly product remains unclear.

**Response**

We agree that the first version of the manuscript does not distinguish enough between the monthly and yearly products. We believe that now, we have added enough motivation for why we are having two separate products (please refer to our response to comment 2). We are also specifying which product is used in each analysis (please refer to our response to comment 25).

**Reviewer Comment 32**

L233: Language like "it is well established" sounds a bit too grand here, especially if both citations in that sentence are from Ito et al.

**Response** We thank the reviewer for noticing this. We agree that language like "it is well established" is not really appropriate in this context. Besides, there was a typo in the citation, and the second one is from Schmidtko et al. We corrected the sentence following the reviewer suggestions as follows:

It is known that the estimates from Ito et al. (2022) tend to be lower compared to other state-of-the-art studies [Schmidtko et al., 2017].

**Reviewer Comment 33**

L236: You're right that your data density is higher - but it is still a sparse dataset, even with emulated data added. It would be good to acknowledge that there may still be some bias (or argue why you think there isn't).

**Response**

We agree with the reviewer on this point. We think that even with the emulated profiles, there is still a bias due to a lack of data, particularly in the earlier years of the product. We corrected the paragraph as follows:

In this context, our methodology mitigates some of these biases by increasing the density of dissolved oxygen data through the incorporation of ML-emulated oxygen estimates.

L245: You should refer again to figure 7 here again, as it matches what you say about PDO phases. Also, based on that and analysis of the figure by eye a reader could agree. To make it clearer you could also compare your results with the PDO Index, for example. If there is a strong correlation this would improve confidence in these results, as there is always the possibly that part of it is noise (or part of it comes from other processes).

**Response**

We thank the reviewer for this comment. as discussed in our response to comment 4, we have further developed the link between ocean oxygen variability and the PDO. Furthermore, and following the reviewer's suggestion, we also modified Figure 8 by adding an inset showing the smoothed PDO index over the same period. This highlights the alternation between positive- and negative-dominated PDO phases and their correspondence with the variability in O2 content.

**Reviewer Comment 35**

L251: I know you already talked about some of this in section 2.4 with the covariance matrix, but a dedicated section on uncertainty estimates should also explicitly address the prediction/emulation error of the machine learning model, measurement uncertainty, and other relevant components. At present, these elements are just contained in the covariance matrix. At minimum, it would be helpful to restate which uncertainty components are included when you write, "As described in Section 2.4,  $\Sigma_a$  is a diagonal matrix, representing the variance at each grid point."

**Response**

We thank the reviewer for this comment regarding the description of the covariance matrix in the uncertainty quantification section. We have revised the first paragraph of the section to explicitly recall the definition of the analysis covariance matrix and to clarify the main factors that influence its computation.

We analyze the uncertainty fields associated with the proposed gridded product. Uncertainty is quantified using the covariance matrix  $\Sigma_a$  of the reconstructed field. As described in Section 2.4,  $\Sigma_a$  is diagonal, with each entry representing the variance of the estimated value at a specific grid point after assimilating the data. This variance reflects the remaining uncertainty at that location, accounting for both the background (prior) variance and the information provided by the observations. The variance at a given grid point decreases as the number of nearby observations increases and as the quality of those observations improves. In our framework, the quality of the gridded observations is defined through the observation error covariance matrix, which assigns lower uncertainty to real dissolved oxygen data than to emulated data. Additionally, the gridding variance accounts for the natural variability of the ocean within a grid cell and for potential disagreement between real and emulated profiles, further reflecting the overall uncertainty at each location.

Figure 8: These maps along with figures C3 and C4 are useful, but it would be good to see time and space together, i.e. is there a seasonal bias in some regions? You should also address that even with the emulated data added, there are spatial gaps in certain decades (figure C3 and C4)

**Response**

We thank the reviewer for this comment. Indeed, some regions are likely affected by seasonal bias. As discussed in our response to Comment 3, we have added a new figure (Fig. D3) in Appendix D that analyzes the seasonal bias across the major ocean basins. As expected, the results confirm the presence of a bias, most notably in the higher latitudes and in the Southern Ocean. However, we find that the inclusion of the emulated data substantially mitigates this bias, especially during the Argo era. This supports the ability of our reconstruction to reduce the impact of seasonal sampling bias—particularly in the monthly product, which explicitly resolves seasonal variability.

**Reviewer Comment 37**

L270: I'm not convinced if you are actually avoiding these additional sources of errors. You are still interpolating; it's just in a different order. Don't get me wrong, I think this approach is useful, but I wouldn't go so far as suggested here. I think there is not enough testing of remaining biases, as it seems like even with the emulated data added the dataset is still sparse. If you think there has been enough testing, it would be good to make that clearer.

**Response**

We agree with the reviewer that we do not necessary reduce uncertainity w.r.t. state of the art ML based products. However, we do have a better control over the different sources of errors and the way we blend them into the optimal interpolation, which results in uncertainty estimates that are more interpretable in terms of standard OI. We modified the paragraph pointed by the reviewer in this sens, as follows:

Independent Interpolation of Oxygen Data: Unlike recent state-of-the-art ML approaches [Sharp et al., 2022, Ito et al., 2024a] that derive gridded oxygen products from interpolated temperature and salinity fields, our method directly interpolates observed and emulated oxygen data using optimal interpolation (OI). This approach ensures better control over the sources of errors present in data, leading to more interpretable uncertainty estimates of the final gridded product.

L273: Regarding the two products: You talked very little about the differences between the two. It is a bit unclear to the reader what the purpose of creating two different ones is. Are there any things one is not representing as well as the other - for example, is the only purpose of the monthly product to look at the seasonality? Or are there any other advantages? Did the smaller e-folding length scale make a difference? It would be great if you could provide more detail regarding your choices and why these had to be separate products. It would also be good to think about why 1 degree was appropriate.

**Response** We thank the reviewer for this comment. We already answered in our answer to comment 2.

**Reviewer Comment 39**

L281: I can see from the plots why you are excited and use the words "outperforms current state-of-the-art products" but I think you haven't shown enough why you are confident in your results.

**Response**

We thank the reviewer for raising this comment. We have reformulated the paragraph to emphasize that it reveals different type of variability.

The resulting product generally agrees with the reference climatology and recent ML-based products in terms of reproducing the spatial variability of dissolved oxygen, and it also aligns with previously published estimates of long-term global deoxygenation, albeit with reduced uncertainty around those estimates. Moreover, the product reveals interdecadal variability that is absent from existing ML-based reconstructions but consistent with numerical model simulations, suggesting that it better captures the underlying physical processes at these scales.

**Reviewer Comment 40**

Figure C2: The upper left figure, not right.

**Response** We thank the reviewer for noticing this typo. It was corrected.

**Reviewer Comment 41**

Figures C3 and C4: I think these are important figures that further show how you add emulated data over different 5-year windows. However, they don't seem to be referenced in the text.

**Response** We thank the reviewer for raising this point. We agree that these figures are important, as they show the improvement in the sampling of ocean dissolved oxygen concentration due to the emulated data. We have added a section that refers to these figures in the data section as follows:

REFERENCES 19

To highlight the impact of adding these emulated profiles on the sampling of dissolved oxygen concentration, we present in Appendix D the spatial coverage of the observed dissolved oxygen data (Figure D4) and that of the emulated oxygen data derived from temperature and salinity profiles (Figure D5), along with an analysis of the sampling bias before and after including the emulated profiles (Figure D3). Overall, we observe a substantial increase in data coverage resulting from the inclusion of the emulated profiles, particularly after the year 2000, which coincides with the deployment of the Argo program.

**References**

- Dudley B Chelton, Michael G Schlax, Roger M Samelson, and Roland A de Szoeke. Global observations of large oceanic eddies. *Geophysical Research Letters*, 34(15), 2007.
- Lijing Cheng, Kevin E Trenberth, John Fasullo, Tim Boyer, John Abraham, and Jiang Zhu. Improved estimates of ocean heat content from 1960 to 2015. *Science Advances*, 3(3):e1601545, 2017.
- Hervé Claustre, Kenneth S Johnson, and Yuichiro Takeshita. Observing the global ocean with biogeochemical-argo. *Annual review of marine science*, 12(1):23–48, 2020.
- Curtis Deutsch, Holger Brix, Taka Ito, Hartmut Frenzel, and LuAnne Thompson. Climate-forced variability of ocean hypoxia. *science*, 333(6040):336–339, 2011.
- Olaf Duteil, Andreas Oschlies, and Claus W Böning. Pacific decadal oscillation and recent oxygen decline in the eastern tropical pacific ocean. *Biogeosciences*, 15(23):7111–7126, 2018.
- Fabienne Gaillard, Thierry Reynaud, Virginie Thierry, Nicolas Kolodziejczyk, and Karina Von Schuckmann. In situ-based reanalysis of the global ocean temperature and salinity with isas: Variability of the heat content and steric height. *Journal of Climate*, 29(4):1305–1323, 2016.
- Marilaure Grégoire, Véronique Garçon, Hernan Garcia, Denise Breitburg, Kirsten Isensee, Andreas Oschlies, Maciej Telszewski, Alexander Barth, Henry C Bittig, Jacob Carstensen, et al. A global ocean oxygen database and atlas for assessing and predicting deoxygenation and ocean health in the open and coastal ocean. *Frontiers in Marine Science*, 8:724913, 2021.
- Nicolas Gruber, Scott C Doney, Steven R Emerson, Denis Gilbert, Taiyo Kobayashi, Arne Körtzinger, Gregory C Johnson, Kenneth S Johnson, Stephen C Riser, and Osvaldo Ulloa. Adding oxygen to argo: Developing a global in situ observatory for ocean deoxygenation and biogeochemistry. 2010.
- JC Hermes, Y Masumoto, LM Beal, Mathew Koll Roxy, Jérôme Vialard, M Andres, H Annamalai, Swadhin Behera, N d'Adamo, T Doi, et al. A sustained ocean observing system in the indian ocean for climate related scientific knowledge and societal needs. *Frontiers in Marine Science*, 6:355, 2019.
- Takamitsu Ito. Optimal interpolation of global dissolved oxygen: 1965–2015, 2022.
- Takamitsu Ito, Matthew C Long, Curtis Deutsch, Shoshiro Minobe, and Daoxun Sun. Mecha-

- nisms of low-frequency oxygen variability in the north pacific. *Global Biogeochemical Cycles*, 33(2):110–124, 2019.
- Takamitsu Ito, Ahron Cervania, Kaylin Cross, Sanika Ainchwar, and Sara Delawalla. Mapping dissolved oxygen concentrations by combining shipboard and argo observations using machine learning algorithms. *Journal of Geophysical Research: Machine Learning and Computation*, 1(3):e2024JH000272, 2024a.
- Takamitsu Ito, Hernan E Garcia, Zhankun Wang, Shoshiro Minobe, Matthew C Long, Just Cebrian, James Reagan, Tim Boyer, Christopher Paver, Courtney Bouchard, et al. Underestimation of multi-decadal global o 2 loss due to an optimal interpolation method. *Biogeosciences*, 21(3):747–759, 2024b.
- Andreas Oschlies, Peter Brandt, Lothar Stramma, and Sunke Schmidtko. Drivers and mechanisms of ocean deoxygenation. *Nature geoscience*, 11(7):467–473, 2018.
- Mathieu A Poupon, Laure Resplandy, Marina Lévy, and Laurent Bopp. Pacific decadal oscillation influences tropical oxygen minimum zone extent and obscures anthropogenic changes. *Geophysical Research Letters*, 50(7):e2022GL102123, 2023.
- Dean Roemmich, Gregory C Johnson, Stephen Riser, Russ Davis, John Gilson, W Brechner Owens, Silvia L Garzoli, Claudia Schmid, and Mark Ignaszewski. The argo program: Observing the global ocean with profiling floats. *Oceanography*, 22(2):34–43, 2009.
- Sunke Schmidtko, Lothar Stramma, and Martin Visbeck. Decline in global oceanic oxygen content during the past five decades. *Nature*, 542(7641):335–339, 2017.
- Jonathan D Sharp, Andrea J Fassbender, Brendan R Carter, Gregory C Johnson, Cristina Schultz, and John P Dunne. Gobai-o 2: temporally and spatially resolved fields of ocean interior dissolved oxygen over nearly two decades. *Earth System Science Data Discussions*, 2022:1–46, 2022.
- Matias Valdenegro-Toro and Daniel Saromo Mori. A deeper look into aleatoric and epistemic uncertainty disentanglement. In 2022 IEEE/CVF Conference on Computer Vision and Pattern Recognition Workshops (CVPRW), pages 1508–1516. IEEE, 2022.